# SLEEP2VEC: UNIFIED CROSS-MODAL ALIGNMENT FOR HETEROGENEOUS NOCTURNAL BIOSIGNALS

**Weixuan Yuan**[1,6,*]**, Zengrui Jin**[2,3,*]**, Yichen Wang**[1,3]**, Donglin Xie**[4]**,**
**Ziyi Ye**[5]**, Chao Zhang**[2,3,†]**, Xuesong Chen**[1,3,†]

[1]Five Seasons Medical, [2]Tsinghua University, [3]Beijing Key Laboratory for Sleep Breathing Disorder,
[4]Peking University, [5]Fudan University, [6]Technical University of Munich
{zrjin, cz277}@tsinghua.edu.cn; chenxuesong@wuji-inc.com

## ABSTRACT

Tasks ranging from sleep staging to clinical diagnosis traditionally rely on standard polysomnography (PSG) devices, bedside monitors and wearable devices, which capture diverse nocturnal biosignals (e.g., EEG, EOG, ECG, $SpO_2$). However, heterogeneity across devices and frequent sensor dropout pose significant challenges for unified modelling of these multimodal signals. We present `sleep2vec`, a foundation model for diverse and incomplete nocturnal biosignals that learns a shared representation via cross-modal alignment. `sleep2vec` is contrastively pre-trained on 42,249 overnight recordings spanning nine modalities using a *Demography, Age, Site & History-aware InfoNCE* objective that incorporates physiological and acquisition metadata (*e.g.*, age, gender, recording site) to dynamically weight negatives and mitigate cohort-specific shortcuts. On downstream sleep staging and clinical outcome assessment, `sleep2vec` consistently outperforms strong baselines and remains robust to any subset of available modalities and sensor dropout. We further characterize, to our knowledge for the first time, scaling laws for nocturnal biosignals with respect to modality diversity and model capacity. Together, these results show that unified cross-modal alignment, coupled with principled scaling, enables label-efficient, general-purpose modelling of real-world nocturnal biosignals.

## 1 INTRODUCTION

Sleep is a central determinant of human health, it shapes cognition, metabolism, cardiovascular function, and mental well-being, and its disruption both signals and drives disease (Irwin, 2015; Mukherjee et al., 2015; Leng et al., 2019; Lim et al., 2023). Sleep is clinically assessed with polysomnography (PSG) (Bloch, 1997; Boulos et al., 2019), which is a gold standard multi-sensor recording that jointly measures neural and ocular electrophysiology, muscle tone, cardiorespiratory dynamics, and oxygen saturation. Outside the clinical facilities, a growing range of bedside monitors and wearable devices captures subsets of these PSG modalities, creating a fragmented landscape across devices and care settings (Paalasmaa et al., 2012; Sadek et al., 2020; Birrer et al., 2024; Yu et al., 2025; Pillai et al., 2025). This reality motivates the question:

" Can cross-modal alignment of nocturnal biosignals enable a unified physiological representation that generalizes robustly across heterogeneous sensor sets in sleep medicine? "

Physiological signal pre-training offers a promising paradigm by learning generalized representations from diverse biosignals with minimal supervision (Thapa et al., 2024; 2025; Pillai et al., 2025; Fox et al., 2025). Yet real-world data bring hard constraints, sensor montages vary across centers and devices, sampling rates differ, entire channels are often missing, and large-scale expert annotation remains costly, making such a foundation both necessary and challenging.

We posit that concurrent nocturnal signals represent multiple perspectives of the same latent physiological state (Rechtschaffen & Kales, 1968; Berry et al., 2012; 2017). A proper alignment of

---

*Equal contribution was made between the first two authors.
†Corresponding author.

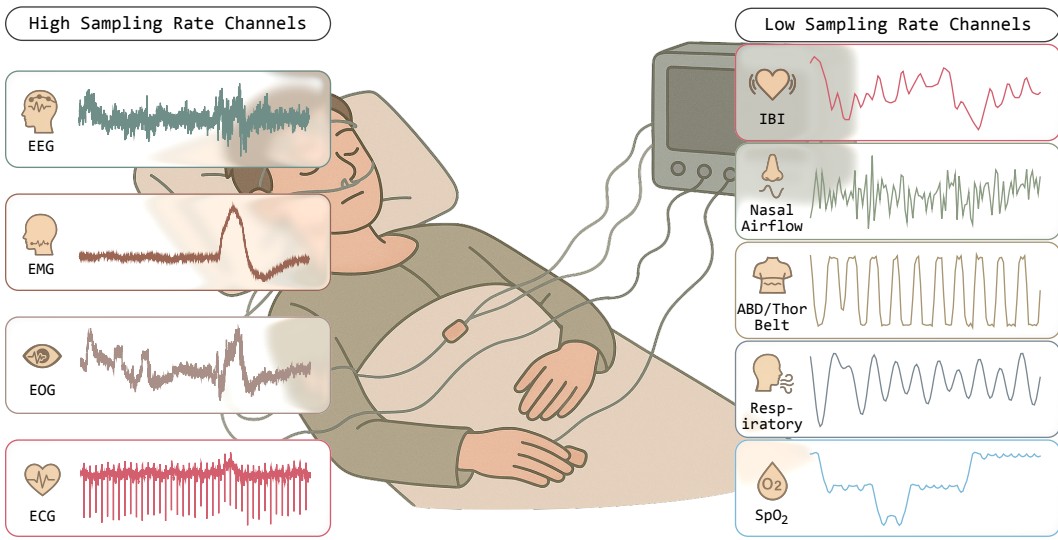

Figure 1: Polysomnography (PSG) captures diverse physiological signals, illustrated here as 30-second segments for each modality. High sampling rate electrophysiological channels include EEG, EMG, EOG, and ECG, while lower sampling rate cardiopulmonary and oximetry channels encompass Nasal Airflow, Abdominal/Thoracic Belt (ABD/Thor Belt), and $SpO_2$. Inter-Beat Interval (IBI) and Respiratory effort (RESP) are interval-derived features computed from ECG and respiratory channels (ABD/Thor Belts when available, otherwise Nasal Airflow), respectively. Together, these concurrent nocturnal signals provide complementary perspectives on a shared latent physiological state, highlighting the multimodal complexity inherent to sleep monitoring.

these heterogeneous views into a unified representation space enables downstream tasks to flexibly operate on arbitrary modalities without retraining specialized pipelines. Such a space must yield modality-agnostic representations robust enough to ensure reliable inference even when modality missing occurs. This leads to a scaling hypothesis, suggesting that increasing modality diversity and model capacity can enrich semantic coverage and regularize modality-specific nuances. Although scaling laws have been extensively studied in language and vision, their implications remain largely unexplored in physiological signal contexts. We therefore propose and evaluate a framework demonstrating predictable benefits of scaling PSG foundation models along both modality and parameter axes, especially for cross-center generalization where variations in sensor montage, demographics, and acquisition protocols are prevalent.

Prior work only partially addresses these needs. Existing models are typically trained for a specific downstream task (Wang et al., 2024; Shen et al., 2024; Carter et al., 2024; Pan et al., 2024; Shen et al., 2024; Lee et al., 2025; Ma et al., 2025; Fox et al., 2025), lacking the generality required of a foundation model capable of supporting multiple tasks. Contrastive pre-training has shown promise on limited sets of physiological, typically one to three channels (*e.g.*, EEG and ECG) (Wang et al., 2024; Mathew et al., 2024; Thapa et al., 2024; Zhou et al., 2025; Thapa et al., 2025), but has not scaled to the full palette of PSG sensors. When more modalities are involved, objectives often prioritize reconstruction (Narayanswamy et al., 2024; Luo et al., 2024; Mathew et al., 2024; Nie et al., 2025) rather than explicit cross-modal alignment. Reconstruction encourages fidelity to modality-specific details but does not enforce that heterogeneous inputs map to a shared semantic manifold. As a result, inference typically assumes access to the same modality set used in training, degrading under realistic sensor missing scenarios. Moreover, systematic analyses of how performance scales with modalities and parameters are scarce.

We address these gaps with `sleep2vec`, a PSG foundation model that aligns heterogeneous nocturnal signals into a unified embedding space. Our framework jointly leverages nine modalities, waveform channels including EEG, EOG, EMG, ECG, Nasal airflow, Abdominal/Thoracic Belt (ABD/Thor Belt) and $SpO_2$; and interval-derived features including Inter-beat Interval (IBI) and Respiratory effort (RESP), from 42,249 nights of physiological recordings. A context-aware InfoNCE objective, explicitly modelling physiological similarity (age, gender, recording center) to

dynamically weight samples, effectively distinguishes hard from easy negatives, mitigating overfitting to dataset-specific nuances.

Our work makes the following contributions:

**(i)** Unified multimodal PSG pre-training: We propose, to our knowledge, the largest scale multimodal contrastive pre-training framework for PSG foundation models, jointly aligning waveform and interval-based modalities, uncovering comprehensive inter-modal physiological correlations.

**(ii)** Scaling law investigation: We systematically explore scaling PSG foundation models along modality diversity and parameter dimensions, demonstrating predictable improvements in cross-cohort generalization with minimal task-specific labels.

**(iii)** Cross-modal training objective: We propose *Demography, Age, Site & History-aware InfoNCE* (DASH-InfoNCE), a context-aware contrastive objective that conditions negative-sample weighting on demographic, age, acquisition-site, and recording-history metadata. This metadata-guided weighting suppresses cohort-specific shortcuts and improves robustness and cross-site generalization across heterogeneous PSG sensor montages.

**(iv)** Comprehensive downstream evaluation: We extensively evaluate sleep2vec on both SHHS and WSC datasets, spanning tasks such as sleep staging, demographic prediction, and diagnostic outcomes, representing the broadest evaluation of a PSG foundation model to date.

## 2 RELATED WORK

**Multimodal alignment for flexible inference.** Contrastive alignment maps heterogeneous inputs into a shared embedding space, enabling zero-shot transfer, retrieval, and robustness to input permutations. In vision–language, CLIP popularized large-scale image–text alignment (Radford et al., 2021), and many-to-one binding across six modalities in ImageBind (Girdhar et al., 2023). *In particular, PSG contains tens of synchronized channels (EEG, EOG, EMG, ECG, Nasal airflow, Respiratory effort, $SpO_2$, etc.), yet prior multimodal alignment works seldom extend beyond EEG-only or a few paired channels, and rarely handle montage shifts at this scale.*

**Self-supervised learning for sleep and PSG data.** Self-supervised learning (SSL) for sleep data has evolved from early approaches targeting task-specific objectives such as sleep staging using fixed PSG montages (Supratak et al., 2017; Perslev et al., 2021), toward broader pre-training frameworks. Recent works emphasize constructing foundational models but typically remain limited by: (i) pre-training strategies narrowly tailored to single downstream tasks or restricted label sets (Fox et al., 2025), (ii) alignment restricted to selected subsets of PSG channels, thus failing to address comprehensive multimodal integration (Fang et al., 2024; Narayanswamy et al., 2024; Luo et al., 2024; Thapa et al., 2024; 2025), or (iii) employing cross-modal generative methods that prioritize modality-specific signal fidelity rather than explicitly aligning heterogeneous modalities (Chen et al., 2024; Nie et al., 2025). *Consequently, cross-modal alignment covering the full PSG spectrum remains largely unexplored, with most pre-training focusing on fixed, small montages.*

**Scaling and generalization.** In language and vision, performance follows predictable trends as model and data scale. Despite rapid progress, systematic studies of scaling laws for physiological time series and PSG remain sparse. Existing PSG SSL studies seldom probe modality-diversity scaling or parameter scaling. *To our knowledge, systematic modality-diversity scaling has not been charted in sleep; existing studies also under-report parameter/data scaling for PSG SSL, leaving open how capability grows with both model size and channel count.*

## 3 METHOD

### 3.1 DATASET AND PREPROCESSING

We leveraged publicly available PSG datasets for pre-training, including the Human Sleep Project (HSP) (Sun et al., 2023) and four cohorts obtained from the National Sleep Research Resource (NSRR) (Zhang et al., 2018): the Sleep Heart Health Study (SHHS) (Quan et al., 1997), Osteoporotic Fractures in Men Study (MrOS) (Blackwell et al., 2011), Multi-Ethnic Study of Atheroscle-

rosis (MESA) (Chen et al., 2015) and Wisconsin Sleep Cohort (WSC) (Young et al., 2009), collectively encompassing multi-center, multi-device acquisitions from diverse demographic populations (age range: 1-109; recording span: 1995-present). Table 4 presents an overview of the five pre-training cohorts, Apnea Positive Pressure Long-term Efficacy Study (APPLES) Quan et al. (2011) is additionally used as an external validation cohort. The five datasets were harmonized into a unified corpus comprising 42,249 overnight recordings from 30,852 subjects, processed through a standardized pipeline to minimize cohort-specific biases and ensure symmetrical handling during batching and evaluation.

A pool of nine PSG channels across cohorts was established as shown in Figure 1: comprising two groups of signals differentiated by sampling rates: higher sampling rate electrophysiological signals, including EEG, EOG, EMG, ECG uniformly resampled to 128 Hz, and lower sampling rate physiological signals, including Nasal airflow, ABD/Thor belt, SpO$_2$, Inter-Beat Interval (IBI) and respiratory signals uniformly resampled to 4 Hz. Only minimal preprocessing is applied to the higher sampling rate signals to preserve raw signal characteristics crucial for downstream physiological interpretation, involving temporal resampling to the target frequency and z-score normalization with cohort-invariant statistics. The IBI channel is derived from ECG R-peak detection, with raw inter-beat intervals cleaned for outliers and artifacts and then linearly interpolated to a continuous 4 Hz sequence. The Respiratory effort reflects breathing cycles extracted from either Nasal Airflow or Abdominal Belt, standardized by band-limiting, and resampling to 4 Hz. Both IBI and Respiratory effort can be captured not only by PSG but also using simpler, low-burden hardware such as ballistocardiography mats or other contactless sensors (Chen et al., 2025).

Participant information is retained when available, including age, gender, and recording site, to facilitate cohort-aware analysis and difficulty estimation during pre-training. Participant-level data partitions are established to prevent data leakage across splits. A dedicated pre-training split (N$_{\text{pre-train}}$=23,934 participants) is exclusively reserved for foundation model learning, while downstream splits follow an 8:1:1 ratio (N$_{\text{train}}$/N$_{\text{val}}$/N$_{\text{test}}$=8,792/1,102/1,116), ensuring no participant overlap and identical modality coverage. For downstream evaluation, sleep staging labels on SHHS and WSC as well as clinical diagnosis labels on SHHS were aligned with the same participant-level splits as in pre-training, ensuring consistency and preventing any data leakage.

## 3.2 MODEL ARCHITECTURE

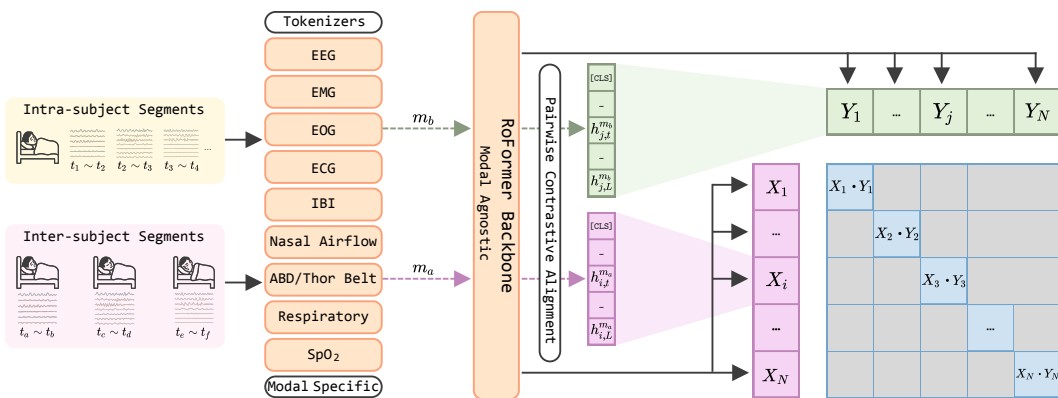

Figure 2: An illustration of the multimodal pre-training framework. Each overnight PSG recording is partitioned into intra-subject segments (different temporal slices from the same individual) and inter-subject segments (slices from different individuals), which are independently tokenized via modality-specific MLP tokenizers. A learnable [CLS] token is prepended to each masked sequence before processing through a modality-agnostic RoFormer backbone. Hidden states from the backbone at each timestep are projected into a shared alignment space, enabling timestep-wise pairwise contrastive alignment across modalities.

A minimalistic tokenizer based on a multi-layer perceptron (MLP) was implemented, comprising two feed-forward layers and a residual connection. The tokenizer maps input 30-second tokens into embeddings of dimension $D$ through an initial linear transformation that projects inputs into an

intermediate hidden representation of dimension $2D$, activated by the SiLU nonlinearity (Elfwing et al., 2018) and regularized using dropout with a 0.1 probability. The choice of 30-second token aligns with the standard epoch duration recommended by the American Academy of Sleep Medicine (AASM) guidelines (Berry et al., 2012; 2017) for polysomnographic analysis. Subsequently, this hidden representation is linearly transformed into the final embedding space (D). In parallel, a residual linear transformation directly maps the inputs into the output embedding dimension, enhancing gradient flow and training stability. A LayerNorm is further applied to the final embedding for normalization. Cross-modal sampling rate differences are resolved by encoding 30-second tokens into tokens of equal embedding dimension using modality-specific tokenizers, each operating directly on the original sampling rates, resulting in temporally aligned embeddings across modalities to be fed into the modality-agnostic backbone.

As illustrated in Figure 2, to maintain stable optimization as the number of modalities increases, each mini-batch is constructed around a single modality pair $(m_a, m_b)$. At the batch level, two modalities are sampled from those available in the batch, and each training sample contributes one aligned view from each selected modality, forming the contrastive pairs used for pre-training. Independent time-step masking is then applied to these paired instances to enhance robustness and mitigate shortcut learning. Each 30-second token has a 15% probability of being replaced by a learnable, modality-specific mask token, after which alignment is conducted exclusively between these masked segments. A dedicated learnable `[CLS]` token is prepended to the sequence, the resulting input is then processed through a modality-agnostic RoFormer backbone (Su et al., 2024). It is important to emphasize that the RoFormer backbone in sleep2vec should be viewed as one concrete instantiation of a generic modality-agnostic sequence encoder rather than a core contribution in isolation. The aim is not to advocate RoFormer as the uniquely optimal architecture for PSG, but to show that pairing a flexible backbone capable of ingesting arbitrary channel subsets with a metadata-aware contrastive alignment objective provides a simple and effective recipe for handling heterogeneous PSG montages. In principle, other Transformer style or state-space sequence encoders could be substituted without changing the overall framework, and we expect the benefits of unified cross-cohort pre-training and metadata-aware alignment to transfer across such choices. The backbone outputs hidden states for each timestep, as well as a global nocturnal representation at the `[CLS]` position. These hidden states are projected into a shared 128-dimensional alignment space via a shared three-layer MLP projection head, enabling the application of a cross-modal contrastive loss at each timestep.

During fine-tuning, both masking and the contrastive learning projection head are removed, and modal configurations remain fixed per downstream task. Task-specific heads directly operate on backbone features. Sequence-level tasks (*e.g.*, sleep staging) use per-time-step hidden states, while aggregate tasks (*e.g.*, gender, age, or clinical diagnosis) rely on the global nocturnal representation from the `[CLS]` position. When multiple modalities are available at inference, their representations are aggregated using simple fusion strategies such as averaging, concatenation, or a small gating module. Specific fusion methods employed per task are detailed in the experimental results section.

### 3.3 CROSS-MODAL ALIGNMENT OBJECTIVE: DASH-INFONCE

During pre-training, each mini-batch contains $B$ paired segments, each of length $L$ timesteps. For segment index $i \in \{1, \ldots, B\}$, time index $t \in \{1, \ldots, L\}$, and modality $m \in \{m_a, m_b\}$, denote $\mathbf{v}_{i,t}^{(m)} \in \mathbb{R}^d$ as the corresponding $d$-dimensional embedding. Given $\widehat{\mathbf{v}}_{i,t}^{(m)}$ as its $\ell_2$ normalized product given by $\widehat{\mathbf{v}}_{i,t}^{(m)} = \mathbf{v}_{i,t}^{(m)} / \|\mathbf{v}_{i,t}^{(m)}\|_2$, the cosine similarity is

$$s_{i,j,t} = \left\langle \widehat{\mathbf{v}}_{i,t}^{(m_a)}, \widehat{\mathbf{v}}_{j,t}^{(m_b)} \right\rangle \in [-1, 1], \qquad i, j \in \{1, \ldots, B\}, \ t \in \{1, \ldots, L\}, \tag{1}$$

where $\langle \cdot, \cdot \rangle$ denotes the dot product. The index mapping $\pi : 1, \ldots, B \to 1, \ldots, B$ specifies the number of paired segments in modality $m_b$ for an anchor in modality $m_a$, where batches are typically aligned such that $\pi(i) = i$. Demographic and acquisition metadata for segment $i$ are denoted by $(a_i, g_i, c_i, u_i)$, where $a_i \in \mathbb{R}_+$, $g_i \in \mathcal{G}$, $c_i \in \mathcal{C}$, and $u_i$ represent age, gender, acquisition site, and the subject-night identifier, respectively. These variables are used solely for weighting and modulation below and never as labels in the learning objective.

### 3.3.1 BASE FORMULATION: TEMPORAL INFONCE

With temperature $\tau > 0$, the baseline timestep InfoNCE loss aligning $m_a$ to $m_b$ is

$$\mathcal{L}_{\text{base}}^{(t)} = \frac{1}{B} \sum_{i=1}^{B} \left[ -\log \frac{\exp\left(s_{i,\pi(i),t}/\tau\right)}{\sum_{j=1}^{B} \exp\left(s_{i,j,t}/\tau\right)} \right]. \tag{2}$$

This objective encourages the similarity between the paired cross-modal embeddings $(i, \pi(i), t)$ to exceed the similarities to all in-batch, same-time candidates $(i, j, t)$ with $j \neq \pi(i)$. The temperature coefficient $\tau$ controls the concentration of the induced softmax distribution.

### 3.3.2 PROPOSED DASH-INFONCE LOSS

A novel DASH-InfoNCE loss that reshapes the negative set by (i) metadata-driven sample weighting and (ii) margin-based modulation of *pseudo-negatives* (*i.e.*, negatives from the same subject-night) is introduced in this section. For anchor $(i, t)$, define

$$\ell_{\text{DASH}}(i, t) = -\log \frac{\exp\left(s_{i,\pi(i),t}/\tau\right)}{\sum_{j=1}^{B} \omega_{i,j} \, \exp\left(\left[s_{i,j,t} - \gamma \, \psi(d_{i,j}, p_{i,j,t})\right]/\tau\right)}, \tag{3}$$

where $\omega_{i,j} \geq 0$ are segment-specific weights satisfying $\sum_{j=1}^{B} \omega_{i,j} = 1$, $\gamma \geq 0$ is a modulation strength, and $\psi(\cdot, \cdot) \geq 0$ reduces the effective logit of designated pseudo-negatives before the softmax. The binary indicator $d_{i,j} \in {0, 1}$ selects which pairs are margin-modulated, with the convention $d_{i,\pi(i)} = 0$ ensuring that positives are not penalized. The optional factor $p_{i,j,t} \in [0, 1]$ encodes time-specific signals, and in our instantiation below, we set $\psi$ to a fixed margin with $p_{i,j,t}$ absorbed into that choice. Relative to Eq. (2), the numerator is unchanged while the denominator concentrates probability mass on demographically similar, presumably harder negatives via $\omega_{i,j}$ and diminishes the competitive strength of same-subject-night negatives through the subtractive margin $\gamma \, \psi$.

### 3.3.3 SAMPLE WEIGHTING MECHANISM

Let $\kappa : \mathbb{R}_+ \times \mathbb{R}_+ \to \mathbb{R}_+$ be a non-negative, symmetric kernel that decreases with the age difference $|a_i - a_j|$. We further define similarity factors for gender and acquisition site as $s_{i,j}^{(g)} \in {\gamma_{\text{same}}, \gamma_{\text{diff}}}$ and $s_{i,j}^{(c)} \in {\delta_{\text{same}}, \delta_{\text{diff}}}$, with $\gamma_{\text{same}} > \gamma_{\text{diff}} \geq 0$ and $\delta_{\text{same}} > \delta_{\text{diff}} \geq 0$, where the value is chosen according to whether $g_i = g_j$ and $c_i = c_j$, respectively.

Given the *pseudo-negative* indicator $h_{i,j} = \mathbb{I}[u_i = u_j \wedge j \neq \pi(i)] \in {0, 1}$, the unnormalized weights are defined as $\alpha_{i,j} = \kappa(a_i, a_j) \, s_{i,j}^{(g)} \, s_{i,j}^{(c)} + \varepsilon \, h_{i,j}$, where $\varepsilon = 10^{-6}$. The normalized weights are computed by

$$\omega_{i,j} = \frac{\alpha_{i,j}}{\sum_{k=1}^{B} \alpha_{i,k}}, \qquad \sum_{j=1}^{B} \omega_{i,j} = 1. \tag{4}$$

This weighting scheme assigns higher values to negatives closely matched by age, gender, and acquisition site. Constant $\varepsilon$ ensures negatives from the same subject-night retain non-zero weights, stabilizing the denominator in Eq. (3) when closely matched demographic negatives are rare.

### 3.3.4 PSEUDO-NEGATIVE MODULATION

We modulate only negatives drawn from the same subject-night. Let $d_{i,j} = h_{i,j}$ and take a margin-only instantiation of $\psi$:

$$\psi(d_{i,j}, p_{i,j,t}) = \begin{cases} m, & d_{i,j} = 1, \\ 0, & d_{i,j} = 0, \end{cases} \qquad m > 0. \tag{5}$$

By combining Eq. (5) and Eq. (3), the fixed margin $\gamma m$ is subtracted from the logits of same-subject-night negatives prior to the softmax. This reduces the tendency to over-penalize semantically close negatives originating from the same subject-night while preserving their presence in the denominator through Eq. (4).

### 3.3.5 FINAL OBJECTIVE

The DASH-InfoNCE objective averages the per-anchor loss Eq. (3) over instances and time:

$$\mathcal{L}^{(t)}_{\text{DASH}} \ = \ \frac{1}{B}\sum_{i=1}^{B}\ell_{\text{DASH}}(i,t), \qquad \mathcal{L}_{\text{DASH}} \ = \ \frac{1}{L}\sum_{t=1}^{L}\mathcal{L}^{(t)}_{\text{DASH}}. \tag{6}$$

Averaging across $t$ enforces alignment at each timestep. Note that every component of Eq. (3)–Eq. (6) depends only on demographic and acquisition metadata $(a_i, g_i, c_i)$ and identifiers $u_i$ via Eq. (4), no downstream task labels are used during pre-training.

### 3.4 FEATURE FUSION

In multimodal physiological tasks, the way modality-specific features are fused/aggregated has a direct impact on performance. A naïve **Concat** strategy (*i.e.* concatenating embeddings before classification) produces high-dimensional, sparse representations that inflate computation and sample complexity, exacerbating overfitting. Conversely, **Mean** aggregation (*i.e.* element-wise averaging) assumes equal informativeness and reliability across channels; in practice, physiological streams differ in SNR and complementary content, so uniform averaging washes out modality-specific cues and is brittle under missing sensors.

To address these limitations, we adopt the **Gating Mechanism**, which introduces learnable scalar weights assigned to each modality. This approach adaptively emphasizes modalities based on their informativeness, dynamically adjusting the contribution of each PSG channel. Consequently, it yields a more expressive, compact, and task-oriented aggregated representation, enabling efficient downstream learning.

## 4 EXPERIMENTS

### 4.1 PRE-TRAINING DIAGNOSTICS: ALIGNMENT & RETRIEVAL

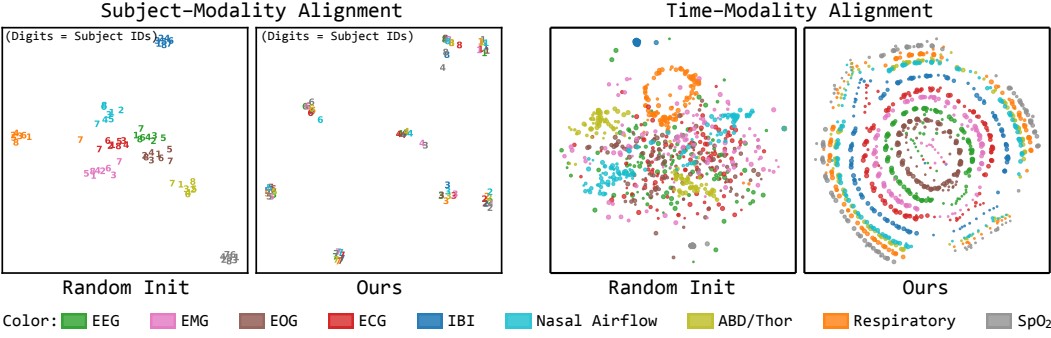

Figure 3: t-SNE visualization of encoder embeddings comparing random initialization and post-pre-training results. **Left Panel (Subject–Modality Alignment):** Visualization of `[CLS]` token embeddings shows that pre-training effectively clusters embeddings from different modalities into distinct, subject-specific groups, indicating aligned subject-level physiological states. **Right Panel (Time–Modality Alignment):** Visualization of timestep-level embeddings, dot sizes indicate temporal ordering (larger → later). Pre-trained embeddings form structured trajectories, contrasting with the scattered distribution observed prior to training.

To assess the effectiveness of multimodal alignment, Figure 3 (left panel) visualizes the `[CLS]` token embeddings using t-SNE both prior to and following pre-training. Initially, embeddings cluster by modality, reflecting intrinsic modality-specific biases and heterogeneous signal characteristics. After pre-training, embeddings from distinct modalities corresponding to the same subject are coherently grouped, indicating improved alignment and preservation of subject-specific structures.

Further analysis of timestep-level embeddings from a random subject (Figure 3, right panel) reveals structured trajectories emerging post-training, indicating an effective modality alignment at a

finer temporal resolution. The coordinated variation in dot sizes across concentric rings emphasizes temporal consistency within the representations. Such consistency is beneficial for downstream sequential tasks like sleep staging, underscoring the practical advantages of the temporally aligned `sleep2vec` embeddings.

## 4.2 DOWNSTREAM FINE-TUNING RESULTS

### 4.2.1 SLEEP STAGING

Table 1: Performance of five-class sleep staging (W/N1/N2/N3/REM) across PSG channel sets and models on **SHHS**. Reported metrics regarding overall performance including Accuracy (Acc., %), Cohen Kappa ($\kappa$), Macro-F1 (MF1, %), Sensitivity (Sens., %) and Specificity (Spec., %). Class-wise F1 (%) is also listed. Baselines reproduced by us for fair comparison are marked with †. Note that these foundation model (FM) baselines were individually pre-trained for each PSG channel subset, whereas `sleep2vec` was pre-trained only once across all modalities. "FULL CHANNELS" refers to the fixed channel configuration that each model is designed for and individually pre-trained on. Underlined numbers indicate the best overall performance within each channel set; **bold numbers** denote the best performance among FMs; **bold-underlined numbers** indicate cases where the FM surpasses specialized models.

| PSG Channel Set | | Overall Performance (↑) | | | | | Class-wise F1 (↑) | | | | |
|---|---|---|---|---|---|---|---|---|---|---|---|
| Inference Subset | Model | Acc. | $\kappa$ | MF1 | Sens. | Spec. | W | N1 | N2 | N3 | REM |
| | *Specialized (non-FM) Model* | | | | | | | | | | |
| | DeepSleepNet (Supratak et al., 2017) | 81.0 | 0.73 | – | – | 73.9 | 85.4 | 40.5 | 82.5 | 79.3 | 81.9 |
| | SleepEEGNet (Mousavi et al., 2019) | 73.9 | 0.65 | – | – | 68.4 | 81.3 | 34.4 | 73.4 | 75.9 | 77.0 |
| | AttnSleep (Eldele et al., 2021) | 84.2 | 0.78 | – | – | 75.3 | 86.7 | 33.2 | 87.1 | 87.1 | 82.1 |
| EEG | XSleepNet1 (Phan et al., 2021) | 87.6 | 0.83 | 80.7 | 79.7 | 96.5 | 91.6 | 51.4 | 88.5 | 85.0 | 88.4 |
| | XSleepNet2 (Phan et al., 2021) | 87.5 | 0.83 | 81.0 | 80.4 | 96.5 | 92.0 | 49.9 | 88.3 | 85.0 | 88.2 |
| | L-SeqSleepNet (Phan et al., 2023) | 87.6 | 0.83 | 80.3 | 79.4 | 96.5 | 92.4 | 48.6 | 88.3 | 83.9 | 88.5 |
| | SleepTransformer (Phan et al., 2022) | 87.7 | 0.83 | 80.1 | 78.7 | 96.5 | 92.2 | 46.1 | 88.3 | 85.2 | 88.6 |
| | *Foundation Model* | | | | | | | | | | |
| | SleepFM (Thapa et al., 2024; 2025) † | 86.3 | 0.81 | 76.3 | 75.3 | **96.1** | **93.2** | 36.6 | 86.3 | 77.3 | 88.1 |
| | sleep2vec | **87.4** | **0.82** | **77.3** | **76.2** | 96.1 | 92.4 | **40.1** | **86.5** | **77.7** | **88.7** |
| | *Specialized (non-FM) Model* | | | | | | | | | | |
| | Sun et al. (2019) † | 71.3 | 0.59 | 57.3 | 56.9 | 91.7 | 85.2 | 4.8 | 70.1 | 49.9 | 76.4 |
| IBI & RESP | Goldammer et al. (2022) † | 77.2 | 0.68 | 63.6 | 62.7 | 93.4 | 88.2 | 15.5 | 76.4 | 55.5 | 82.4 |
| | *Foundation Model* | | | | | | | | | | |
| | SleepFM (Thapa et al., 2024; 2025) † | 79.7 | 0.71 | 65.7 | 65.4 | 94.2 | 90.4 | 12.9 | 78.4 | **61.7** | 84.8 |
| | SleepFounder (Nie et al., 2025) † | 80.9 | 0.73 | **68.3** | **67.0** | 94.5 | **91.3** | **22.3** | 80.0 | 61.1 | **85.9** |
| | sleep2vec | **83.0** | **0.75** | 65.9 | 65.8 | **95.1** | 86.6 | 5.3 | **80.3** | 60.9 | 84.9 |
| | *Foundation Model* | | | | | | | | | | |
| ECG & ABD | SleepFM (Thapa et al., 2024; 2025) † | 77.9 | 0.68 | 62.7 | 62.7 | 93.6 | 88.4 | 6.6 | 76.9 | 60.9 | 80.4 |
| | sleep2vec | **82.7** | **0.75** | **65.6** | **65.2** | **95.0** | **92.6** | **6.2** | **80.6** | **62.4** | **86.1** |
| | *Specialized (non-FM) Model* | | | | | | | | | | |
| | SeqSleepNet (Phan et al., 2019) | 87.2 | 0.82 | 80.2 | 78.7 | 96.3 | 91.8 | 49.1 | 88.2 | 83.5 | 88.2 |
| EEG & EOG & EMG | XSleepNet1 (Phan et al., 2021) | 89.1 | 0.85 | 82.3 | 81.2 | 96.9 | – | – | – | – | – |
| | XSleepNet2 (Phan et al., 2021) | 89.1 | 0.85 | 82.2 | 81.4 | 96.9 | – | – | – | – | – |
| | Olesen et al. (2021) | 85.8 | 0.79 | – | – | – | – | – | – | – | – |
| | *Foundation Model* | | | | | | | | | | |
| | SleepFM (Thapa et al., 2024; 2025) † | 87.0 | 0.82 | 78.0 | 77.8 | 96.4 | 93.6 | **40.7** | 86.8 | 77.8 | **90.9** |
| | sleep2vec | **88.3** | **0.83** | **78.7** | **77.9** | **96.8** | **94.5** | 40.6 | **87.8** | **79.8** | 89.0 |
| | *Foundation Model* | | | | | | | | | | |
| | SleepFM (Thapa et al., 2024; 2025) † | 86.7 | 0.81 | 77.3 | 76.9 | 96.3 | 93.4 | 39.2 | 86.7 | 77.1 | 90.3 |
| FULL CHANNELS | PFTSleep (Fox et al., 2025) | 87.7 | 0.83 | **80.8** | **82.3** | 96.7 | 93.3 | **48.6** | 87.8 | **82.7** | **91.5** |
| | sleep2vec (InfoNCE) | 88.4 | **0.84** | 78.6 | 77.9 | 96.8 | 94.7 | 39.8 | 87.9 | 80.0 | 90.8 |
| | sleep2vec | **88.6** | **0.84** | 79.5 | 78.4 | **96.8** | **94.8** | 44.1 | **88.2** | 79.2 | 91.2 |

We first assess the quality of the learned representations on sleep staging. Experiments are performed on the SHHS and WSC datasets, and the results are presented in Tables 1 and 15, respectively. Several trends can be observed from Table 1:

**(i)** There remains a very limited number of comprehensive works on PSG data, as the majority of existing methods focus narrowly on single-channel EEG or small subsets of physiological signals. Specialized methods typically achieve top performance across available channel sets, setting a challenging baseline for foundation models.

**(ii)** Foundation models generally exhibit lower performance compared to specialized sleep staging approaches optimized specifically for sleep data. This is evident in EEG-only scenarios, where spe-

cialized models consistently hold slight edges in overall metrics compared to baseline FM SleepFM (Acc. 86.3%) and sleep2vec (Acc. 87.4%). However, the gap is marginal, with sleep2vec nearly matching specialized models in certain metrics ($\kappa$ of 0.82 *vs.* 0.83).

**(iii)** sleep2vec consistently outperforms baseline foundation models across all PSG channel subsets. Notable improvements appear in configurations such as "IBI & RESP", where sleep2vec exceeds baseline FMs (Acc.: 83.0% *vs.* SleepFM 79.7% and SleepFounder 80.9%). sleep2vec can also achieve performance comparable to, and in certain cases surpassing, specialized models.

**(iv)** Increasing modality diversity appears beneficial, with sleep2vec consistently demonstrating performance gains when additional physiological signals are included. This trend highlights the scalability and utility of incorporating diverse modalities into foundational model frameworks, further underscoring the capability of sleep2vec to effectively leverage multimodal physiological signals.

To further assess cross-cohort generalization, we evaluate models fine-tuned on SHHS directly on the APPLES cohort, which is unseen during both pre-training and fine-tuning. As shown in Table 2, sleep2vec preserves strong robustness under distribution shift and consistently outperforms baseline methods.

Table 2: Cross-cohort evaluation of five-class sleep staging (W/N1/N2/N3/REM) across PSG channel sets and models on **unseen APPLES**. Models are fine-tuned on SHHS without seeing any data from APPLES during both pre-training and fine-tuning. "FULL CHANNELS" refers to the fixed channel configuration that each model is designed for and individually pre-trained on. Underlined numbers indicate the best overall performance within each channel set; **bold numbers** denote the best performance among FMs; **bold-underlined numbers** indicate cases where the FM surpasses specialized models.

| PSG Channel Set | | Overall Performance (↑) | | | | | Class-wise F1 (↑) | | | | |
|---|---|---|---|---|---|---|---|---|---|---|---|
| Inference Subset | Model | Acc. | $\kappa$ | MF1 | Sens. | Spec. | W | N1 | N2 | N3 | REM |
| IBI & RESP | *Specialized (non-FM) Model* | | | | | | | | | | |
| | (Sun et al., 2019) † | 63.6 | 0.46 | 48.8 | 56.4 | 89.0 | 78.6 | 1.7 | 68.5 | 20.2 | 75.2 |
| | (Goldammer et al., 2022) † | 67.7 | 0.53 | 53.1 | 61.2 | 90.4 | 79.9 | 7.5 | 72.6 | 25.8 | 79.5 |
| | *Foundation Model* | | | | | | | | | | |
| | SleepFM (Thapa et al., 2024; 2025) † | 69.1 | 0.55 | 54.3 | 65.2 | 91.0 | 82.2 | 4.6 | 73.5 | 28.1 | 82.8 |
| | SleepFounder (Nie et al., 2025) † | 68.8 | 0.55 | 55.6 | 67.3 | 91.0 | 85.5 | 8.8 | 71.9 | 27.2 | 84.6 |
| | sleep2vec (InfoNCE) | 71.5 | 0.59 | 56.5 | 66.7 | 91.7 | **86.8** | 7.9 | 74.1 | 29.5 | 84.1 |
| | sleep2vec | **73.2** | **0.61** | **57.8** | 66.2 | **92.1** | 86.5 | **10.2** | **76.5** | **31.5** | 84.2 |
| FULL CHANNELS | *Foundation Model* | | | | | | | | | | |
| | SleepFM (Thapa et al., 2024; 2025) † | 71.4 | 0.59 | 60.0 | 58.9 | 91.6 | 85.5 | 24.5 | 75.8 | 32.0 | 81.5 |
| | sleep2vec (InfoNCE) | 76.8 | 0.67 | 63.5 | 73.3 | 93.4 | **90.5** | 24.0 | 79.5 | 35.5 | 88.1 |
| | sleep2vec | **78.4** | **0.69** | **65.2** | 72.0 | **93.8** | 89.6 | **27.3** | **81.8** | **39.0** | **88.2** |

### 4.2.2 LEAVE-ONE-OUT ANALYSIS

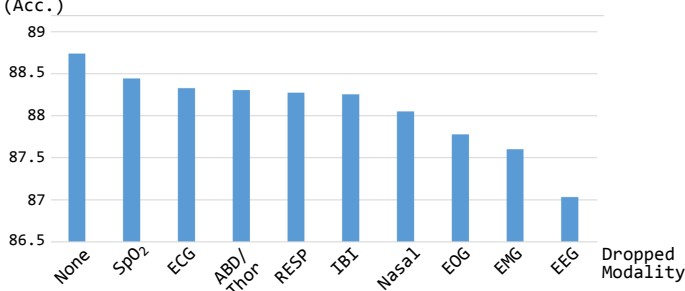

Figure 4: Leave-one-out analysis on the SHHS sleep staging task. Each bar represents model accuracy when one of the nine modalities is excluded during both pre-training and fine-tuning. The observed drop in accuracy relative to the full channels baseline (labeled "None") reflects the contribution and relative importance of each individual modality to the overall model performance.

To further examine the role of individual modalities, we performed a leave-one-out (LOO) study, where one modality was excluded during both pre-training and fine-tuning. Using the same setup

as in Section 4.2.1, we evaluated the model on the SHHS dataset. As shown in Figure 4, excluding modalities such as EEG or IBI leads to a substantial accuracy drop, while others (e.g., SpO$_2$, EOG) have relatively minor effects.

### 4.2.3 CLINICAL DISEASE PREDICTION AND MODALITY SCALING

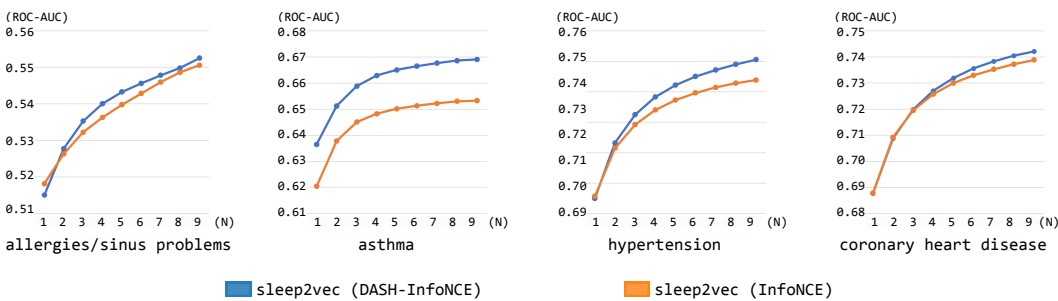

Figure 5: ROC-AUC scores for disease prediction tasks using varying numbers of modalities ($N$) on the SHHS dataset. Results are averaged across all possible modality combinations of size $N$

For clinical evaluation, four prevalent and clinically significant conditions are selected from the SHHS dataset, including *allergies/sinus problems*, *asthma*, *hypertension* and *coronary heart disease*, as shown in Figure 5. These conditions span two major physiological systems directly monitored by PSG, the respiratory system and the cardiovascular system. By including these diverse clinical outcomes, we explicitly test whether cross-modal embeddings generalize robustly across organ systems and sensor subsets.

Specifically, for a given number of modalities $N$, we enumerate all possible modality combinations, build corresponding ensemble models, and report the average ROC-AUC. Results presented in Figure 5 demonstrate: **(i)** clear modality-scaling effects, as performance consistently improves as more modalities are incorporated, suggesting a robust scaling law across clinical prediction tasks; **(ii)** the proposed DASH-InfoNCE loss consistently outperforms the standard InfoNCE baseline, indicating its effectiveness in harnessing richer inter-modal physiological correlations. This performance advantage of DASH-InfoNCE becomes increasingly pronounced with additional modalities, underscoring its efficacy in large-scale multimodal pre-training scenarios.

## 5 CONCLUSION

In this work, we introduced `sleep2vec`, a foundation model aligning multimodal polysomnography (PSG) signals into a unified embedding space for robust physiological representation learning. Leveraging over 42,000 overnight recordings and our novel DASH-InfoNCE loss, which accounts for demographic, age, site, and history variations, we demonstrated significant performance improvements on sleep staging and clinical prediction tasks. Experiments confirmed `sleep2vec`'s robustness to incomplete sensor data and revealed clear scaling laws with increased modality diversity and larger model sizes. Our results establish `sleep2vec` as a scalable and versatile tool, enabling generalized physiological monitoring and clinical decision support in sleep medicine.

## 6 ETHICS STATEMENT

The datasets employed consist of anonymized PSG recordings from publicly available sources. Ethical approval and informed consent for the original data collection were secured by the institutions responsible for the individual studies. All subject identifiers were removed prior to dataset acquisition, ensuring complete anonymization and protecting participants' privacy.

## 7 REPRODUCIBILITY STATEMENT

To ensure the reproducibility of our research, we provide the following details. A comprehensive description of our data processing pipeline is provided in Section 3.1. Details of the datasets involved and the training configurations for the proposed model are presented in Appendices A.2 and A.3, respectively. Furthermore, the specific configuration used for fine-tuning is provided in Appendix A.7.

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

# A APPENDIX

## A.1 PHYSIOLOGICAL SIGNALS IN POLYSOMNOGRAPHY (PSG)

Table 3: Polysomnography (PSG) channels and derived interval-based features used in this study. High sampling rate electrophysiological channels include EEG, EMG, EOG, and ECG; lower sampling rate cardiopulmonary and oximetry channels encompass Nasal airflow, Abdominal/Thoracic belt (ABD/Thor belt), and $SpO_2$. Respiratory effort (RESP) and Inter-Beat Interval (IBI) are interval-derived features, obtained from respiratory channels (ABD/Thor belt if available, otherwise Nasal airflow) and ECG channels, respectively, and can also be measured via wearable devices. *Sampling rate ranges summarize AASM Berry et al. (2012; 2017) minimum recommended digital sampling rates, actual device settings may vary.*

| Physiological Signals | Typical Placement | Sampling (Hz) | Common Usage |
|---|---|---|---|
| ***PSG Channel*** | | | |
| EEG | Scalp | 200–500 | Detecting sleep stages, brain activity patterns, and brief arousals |
| EOG | Around the eyes | 200–500 | Identifying eye movements, especially for REM sleep detection and stage transitions |
| Chin EMG | Under the chin | 200–500 | Measuring muscle tone, useful for distinguishing REM and detecting disorders such as bruxism |
| ECG | Chest leads | 200–500 | Heart activity and variability (HR/HRV), used to study arousals and cardiorespiratory patterns |
| Nasal airflow | Under the nose | 25–100 | Detecting apnoeas/hypopnoeas and breathing irregularities |
| ABD/Thor belt | Around abdomen & chest | 25–100 | Tracking breathing effort, helping to classify types of sleep-disordered breathing |
| $SpO_2$ | Finger probe | 10–25 | Monitoring blood oxygen drops during breathing events, used to measure severity |
| ***Interval-derived Feature*** | | | |
| Respiratory effort (RESP) | – | 4–10 | Breath-to-breath timing, used for variability analysis and detecting abnormal breathing cycles |
| Inter-Beat Interval (IBI) | – | 4–10 | Beat-to-beat timing, used to compute HRV and study autonomic regulation during sleep |

Polysomnography (PSG) is a comprehensive overnight test performed using a polysomnograph that records multiple physiological signals during sleep, including brain activity, eye movements, muscle tone, heart rhythm, breathing patterns, and oxygen levels. An illustration of a subject wearing PSG device for nocturnal sleep recording is presented in Figure 1. It is used in clinical and research settings to diagnose and study sleep disorders, including but not limited to sleep apnea, narcolepsy, and insomnia, by providing an objective assessment of sleep stages and abnormalities.

## A.2 OVERVIEW OF DATASETS

Table 4: Overview of the PSG datasets used in this study.

| Dataset | Age Span[1] | Duration | Recording Span | # Recordings | # Subjects | Total Hours |
|---|---|---|---|---|---|---|
| Sleep Heart Health Study (SHHS) (Quan et al., 1997) | 39-90 | 8.9 ± 1.1 | 1995-2003 | 8,440 | 5,795 | 75,431 |
| Wisconsin Sleep Cohort (WSC) (Young et al., 2009) | 37-85 | 8.0 ± 0.8 | 2000-2015 | 2,570 | 1,123 | 20,520 |
| Osteoporotic Fractures in Men Study (MrOS) (Blackwell et al., 2011) | 67-90 | 11.5 ± 2.3 | 2000-2005 | 3,930 | 2,905 | 45,110 |
| Multi-Ethnic Study of Atherosclerosis (MESA) (Chen et al., 2015) | 54-94 | 10.6 ± 1.6 | 2010-2012 | 2,056 | 2,056 | 21,745 |
| Human Sleep Project (HSP) (Sun et al., 2023) | 1-109 | 7.6 ± 1.1 | 2007-present | 25,253 | 18,973 | 190,732 |
| Apnea Positive Pressure Long-term Efficacy Study (APPLES) (Quan et al., 2011) | 18-83 | 8.2 ± 1.2 | 2003-2004 | 1,096 | 1,096 | 8,955 |

[1]In SHHS and MrOS, ages greater than 90 are top-coded and recorded as 90.

Table 4 compiles six large publicly available PSG cohorts spanning children to older-adult populations (ages 1–109) and nearly three decades of acquisition (1995–present). As indicated by the Duration column in Table 4, all recordings correspond to full-night PSG studies. The corpus of training data comprises 42,249 overnight recordings from 30,852 subjects across these five cohorts (SHHS, WSC, MrOS, MESA, and HSP). In addition, the APPLES cohort, consisting of 1,096 recordings, is used as an external validation cohort. HSP contributes the broadest age range and largest share of data, while SHHS, WSC, MrOS, MESA and APPLES provide well-characterized adult cohorts. The diversity in demographics and collection periods enables robust pre-training and evaluation under heterogeneous sensors and montages.

### A.3 PRE-TRAINING CONFIGURATIONS

During pre-training, we ensured consistency by using a fixed batch size of 320 across all models. Including the prepended [CLS] token, the maximum sequence length was capped at $L = 121$. For contrastive learning, the temperature parameter was set to $\tau = 0.2$. Optimization was performed using AdamW (learning rate $5 \times 10^{-5}$, $\beta = (0.9, 0.95)$, $\epsilon = 10^{-8}$, and weight decay of 0.01 for non-normalization weights), with a linear warmup over 3% of steps followed by cosine decay.

Table 5: Configurations of the `sleep2vec` model across different sizes.

| Configuration | Small | Medium | Large |
|---|---|---|---|
| Number of parameters | 63.5M | 133.7M | 238.2M |
| Hidden dimension | 512 | 768 | 1024 |
| Number of layers | 8 | 12 | 16 |
| Attention heads | 16 | 16 | 16 |
| Segment duration (pre-training) | 1 hour | 1 hour | 1 hour |
| Segment duration (fine-tuning) | Whole night | Whole night | Whole night |

The architectural hyper-parameters are adjusted to yield `sleep2vec` variants with varying numbers of parameters, as detailed in Table 5. Unless otherwise stated, all experiments were conducted using the `sleep2vec`$_\text{medium}$ variant.

For the sample-weighting mechanism introduced in Section 3.3.3, we employed a Laplace kernel

$$\kappa(a_i, a_j) = \exp\left(-\frac{|a_i - a_j|}{\sigma_\text{age}}\right),$$

with bandwidth $\sigma_\text{age} = 20.0$. This choice reflects clinical observations that sleep physiology and sleep-disordered breathing vary gradually with age rather than abruptly. A 20-year scale captures meaningful across-lifespan differences without over-penalizing small age gaps (Ohayon et al., 2004; Li et al., 2022).

Gender coefficients are set to $\gamma_\text{same} = 1.0$ and $\gamma_\text{diff} = 0.8$, acknowledging sex differences in sleep architecture and in the prevalence/severity of sleep-disordered breathing that are present but not dominant at the individual-record level (Peppard et al., 2013). The site coefficients were set to $\delta_\text{same} = 1.3$ and $\delta_\text{diff} = 0.8$ to account for systematic inter-site variation (device, montage, scoring protocol) that is frequently larger than gender effects in multi-center cohorts (Rosenberg & Van Hout, 2013; Kuna et al., 2013). We used $\varepsilon = 10^{-6}$ for numerical stability.

Finally, a fixed margin term $\gamma m = 0.1$ (Eq. 4 and Eq. 5 in Section 3) was applied when modulating pseudo-negatives from the same subject-night, reflecting the high correlation of repeated segments within a recording and discouraging them from being treated as fully independent negatives.

Each pre-training run used two high-memory GPUs, the largest configuration trained for up to 48 hours.

We intentionally avoid any cross-modal reconstruction objective during pre-training. `sleep2vec` is trained solely with the InfoNCE and DASH-InfoNCE losses described above, which encourage alignment between heterogeneous PSG montages and associated metadata without requiring an explicit generative decoder. In preliminary experiments, a variant that replaced the contrastive objective with a generic cross-modal reconstruction module was implemented. Under matched data

and compute budgets, this reconstruction-based variant was substantially harder to optimize and frequently failed to converge to competitive solutions. These observations, combined with the additional computational overhead of large reconstruction decoders, motivated our design choice to focus on contrastive alignment as a more stable and scalable route to robust missing-modality generalization.

## A.4  ABLATION STUDY

### A.4.1  ABLATION OF FEATURE FUSION STRATEGIES

Table 6: Ablation study of different feature-fusion strategies and their impact on five-class sleep-staging performance (W/N1/N2/N3/REM). The evaluated model is the medium-sized sleep2vec variant, fine-tuned on SHHS and evaluated on **unseen APPLES**. **Bold numbers** denote the best performance among FMs.

| Feature Fusion | Overall Performance ($\uparrow$) | | | | | Class-wise F1 ($\uparrow$) | | | | |
|---|---|---|---|---|---|---|---|---|---|---|
| | Acc. | $\kappa$ | MF1 | Sens. | Spec. | Wake | N1 | N2 | N3 | REM |
| Concatenation | 76.9 | 0.66 | 63.1 | 71.8 | 93.3 | 89.9 | 21.0 | 79.9 | 37.3 | 87.4 |
| Mean | 78.1 | 0.68 | 64.7 | 71.9 | 93.7 | 89.3 | 26.2 | 81.5 | 38.4 | 88.1 |
| Gating | **78.4** | **0.69** | **65.2** | **72.0** | **93.8** | **89.6** | **27.3** | **81.8** | **39.0** | **88.2** |

To further assess the influence of different feature-fusion strategies, we perform an ablation study comparing the three representative designs incorporated in our framework: Concatenation, Mean and the adopted Gating mechanism.

In practice, Concatenation rapidly becomes computationally prohibitive as the number of modalities increases, since it expands the hidden representation dimensionality and consequently inflates both the parameter count and VRAM usage of subsequent layers. Additionally, it does not provide measurable performance benefits over the lightweight alternatives and is therefore not used as our default fusion approach.

Our analysis thus focuses on Mean and Gating, two scalable and computationally efficient paradigms. Across representative downstream sleep staging tasks, both strategies achieve competitive performance. Nonetheless, the Gating mechanism consistently yields small but robust improvements over Mean, and further offers enhanced interpretability through modality-specific gating coefficients that quantify the contribution of each input signal.

The results of this ablation study are reported in Table 6. Collectively, these findings justify our choice of Gating as the default fusion strategy, as it provides a balanced combination of scalability, empirical performance and interpretability.

### A.4.2  ABLATION OF METADATA COMPONENTS IN DASH-INFONCE

To isolate the contribution of each metadata component within DASH-InfoNCE, an additional ablation study was conducted to examine the model's generalization behavior under distribution shift. Specifically, we evaluate the impact of incorporating individual metadata, including age, gender, and recording site, on performance in the unseen APPLES cohort.

For this analysis, models are pre-trained with only one metadata enabled at a time, followed by fine-tuning on SHHS and direct evaluation on APPLES without any further adaptation. Downstream sleep staging performance as well as cross-modal retrieval accuracy (Recall@1) are both reported, quantifying the quality of modality alignment in the shared embedding space.

Results presented in Table 7 and Table 8 suggest that activating any single metadata consistently improves performance over the vanilla InfoNCE baseline on the unseen cohort, either through higher Macro-F1 (MF1) or improved retrieval alignment. Combining all three metadata factors in the full DASH-InfoNCE formulation provides the strongest overall performance.

These findings demonstrate that age, gender and site information contribute complementary signals that enhance robustness under distribution shift. Together, they strengthen cross-modal alignment

Table 7: Ablation study of metadata-aware contrastive objectives. Four contrastive formulations are compared during pre-training: (i) vanilla InfoNCE, (ii) single-metadata–aware variants that incorporate one metadata factor at a time (Age-aware, Gender-aware, Site-aware InfoNCE), and (iii) the proposed DASH-InfoNCE. All medium-sized sleep2vec models are pre-trained on the full multimodal corpus and subsequently fine-tuned and evaluated on **SHHS** for five-class sleep staging. "Retrieval Acc." corresponds to recall@1 in a cross-modal retrieval task, given a query embedding from one modality, the model must retrieve the correctly paired PSG segment from a pool of candidates drawn from other modalities, and Retrieval Acc. is the fraction of queries for which the true pair is ranked first. "Retrieval Acc." is computed on the fixed validation set from the pre-training corpus and is independent of the downstream fine-tuning and evaluation cohort. **Bold numbers** denote the best performance among FMs.

| Method | Overall Performance (↑) | | | | | Class-wise F1 (↑) | | | | | Retrieval Acc. (↑) |
|---|---|---|---|---|---|---|---|---|---|---|---|
| | Acc. | $\kappa$ | MF1 | Sens. | Spec. | Wake | N1 | N2 | N3 | REM | |
| Vanilla InfoNCE | 88.4 | **0.84** | 78.6 | 77.9 | **96.8** | 94.7 | 39.8 | 87.9 | **80.0** | 90.8 | 0.351 |
| Age-aware | 88.3 | 0.83 | 78.7 | 77.2 | 96.7 | 94.6 | 41.6 | 88.0 | 78.1 | 91.0 | 0.355 |
| Gender-aware | 88.5 | **0.84** | 79.2 | 78.1 | **96.8** | 94.7 | 43.0 | 88.1 | 79.3 | 91.1 | 0.356 |
| Site-aware | 88.1 | 0.83 | 78.0 | 75.9 | 96.6 | 94.6 | 39.9 | 87.8 | 76.8 | 90.9 | 0.363 |
| DASH-InfoNCE | **88.6** | **0.84** | **79.5** | **78.4** | **96.8** | **94.8** | **44.1** | **88.2** | 79.2 | **91.2** | **0.368** |

Table 8: Ablation study of metadata-aware contrastive objectives on the **unseen APPLES dataset**. All experimental setups, training protocols, and evaluation metrics follow those reported in Table 7, with retrieval accuracy ("Retrieval Acc.") computed on the same validation set. All medium-sized sleep2vec models are pre-trained on the full multimodal corpus and evaluated on **unseen APPLES** for five-class sleep staging. **Bold numbers** denote the best performance among FMs.

| Method | Overall Performance (↑) | | | | | Class-wise F1 (↑) | | | | | Retrieval Acc. (↑) |
|---|---|---|---|---|---|---|---|---|---|---|---|
| | Acc. | $\kappa$ | MF1 | Sens. | Spec. | Wake | N1 | N2 | N3 | REM | |
| Vanilla InfoNCE | 76.8 | 0.67 | 63.5 | **73.3** | 93.4 | **90.5** | 24.0 | 79.5 | 35.5 | 88.1 | 0.351 |
| Age-aware | 78.3 | 0.68 | 64.9 | 72.0 | 93.7 | 89.5 | 26.6 | 81.7 | 39.0 | 88.0 | 0.355 |
| Gender-aware | 78.3 | **0.69** | **65.5** | 72.4 | 93.7 | 89.7 | **29.1** | 81.7 | **39.2** | 87.8 | 0.356 |
| Site-aware | 78.1 | 0.68 | 64.8 | 72.1 | 93.6 | 90.3 | 26.2 | 81.2 | 38.4 | 87.8 | 0.363 |
| DASH-InfoNCE | **78.4** | **0.69** | 65.2 | 72.0 | **93.8** | 89.6 | 27.3 | **81.8** | 39.0 | **88.2** | **0.368** |

and cross-cohort generalization, highlighting DASH-InfoNCE as an effective strategy for improving model stability and transferability in unseen clinical cohorts.

### A.4.3 Ablation of Masking Strategies

Table 9: Ablation study on the effect of masking ratios during pre-training. Other experimental setups, training protocols, and evaluation metrics follow those reported in Table 7, with retrieval accuracy ("Retrieval Acc.") computed on the same validation set. All medium-sized sleep2vec models are fine-tuned and evaluated on **SHHS** for five-class sleep staging. **Bold numbers** denote the best performance among FMs.

| Mask Ratio | Overall Performance (↑) | | | | | Class-wise F1 (↑) | | | | | Retrieval Acc. (↑) |
|---|---|---|---|---|---|---|---|---|---|---|---|
| | Acc. | $\kappa$ | MF1 | Sens. | Spec. | Wake | N1 | N2 | N3 | REM | |
| 0% | 88.5 | **0.84** | 78.8 | 77.2 | 96.7 | 94.7 | 41.9 | **88.2** | 78.2 | 91.1 | **0.403** |
| 15% | **88.6** | **0.84** | **79.5** | **78.4** | **96.8** | **94.8** | **44.1** | **88.2** | **79.2** | **91.2** | 0.368 |
| 30% | 88.4 | 0.83 | 78.8 | 77.2 | 96.7 | 94.7 | 41.6 | 88.1 | 78.6 | 91.1 | 0.281 |

The role of masking strength during contrastive pre-training is also investigated to assess how different corruption levels affect the robustness of the learned representations. Specifically, downstream sleep staging performance of three masking ratios (0%, 15% and 30%) is reported in Table 9 (in-domain SHHS) and Table 10 (cross-cohort APPLES).

Table 10: Ablation study on the effect of masking ratios during pre-training. All experimental setups, training protocols, and evaluation metrics follow those reported in Table 7, with retrieval accuracy ("Retrieval Acc.") computed on the same validation set. All medium-sized sleep2vec models are fine-tuned on **SHHS** and evaluated on **unseen APPLES** for five-class sleep staging. **Bold numbers** denote the best performance among FMs.

| Mask Ratio | Overall Performance (↑) | | | | | Class-wise F1 (↑) | | | | | Retrieval Acc. (↑) |
| | Acc. | $\kappa$ | MF1 | Sens. | Spec. | Wake | N1 | N2 | N3 | REM | |
|---|---|---|---|---|---|---|---|---|---|---|---|
| 0% | 77.9 | 0.68 | 64.0 | 71.4 | 93.6 | 89.4 | 23.2 | 81.5 | 38.6 | 87.4 | **0.403** |
| 15% | **78.4** | **0.69** | **65.2** | 72.0 | **93.8** | 89.6 | **27.3** | **81.8** | **39.0** | **88.2** | 0.368 |
| 30% | 77.5 | 0.67 | 63.4 | **72.2** | 93.5 | **89.8** | 20.0 | 80.8 | 38.7 | 87.9 | 0.281 |

A moderate masking ratio of 15% yields the most favorable balance between representational robustness and downstream accuracy. Compared to the no-masking condition, moderate masking leads to consistent improvements in Macro-F1 (MF1) and class-wise metrics across both evaluation settings. In contrast, increasing the masking ratio to 30% provides no additional generalization benefits, suggesting that overly aggressive masking may overly corrupt physiologically meaningful temporal structure. These results indicate that moderate masking functions as an effective regularizer during contrastive physiological pre-training.

We additionally note that retrieval accuracy is highest under the 0% masking configuration. This trend is likely driven by the closer match between the training objective and the retrieval evaluation when no corruption is applied, rather than reflecting superior generalization. Retrieval accuracy should therefore be interpreted jointly with downstream task performance when comparing masking strategies.

## A.5 FURTHER INVESTIGATION OF ABLATION STUDY

### A.5.1 SCALING LAW OF FOUNDATION MODEL PARAMETERS

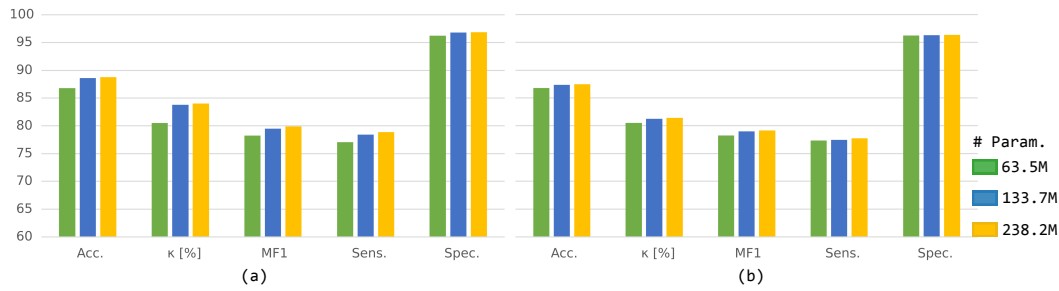

Figure 6: Performance comparison across varying model sizes (63.5M, 133.7M and 238.2M parameters) for sleep staging on **(a)** SHHS and **(b)** WSC datasets. Results demonstrate a clear scaling law, where increasing the number of parameters consistently improves Accuracy (Acc.), Cohen's Kappa ($\kappa$), Macro-F1 (MF1), Sensitivity (Sens.) and Specificity (Spec.), underscoring the effectiveness of scaling physiological foundation models in capturing complex sleep dynamics.

The scaling behavior of `sleep2vec` concerning model parameters is presented in Figure 6. The performance across two benchmark datasets, SHHS **(a)** and WSC **(b)**, consistently improves as the number of parameters increases from 63.5M to 238.2M. This improvement is evident in key metrics such as Accuracy (Acc.), Cohen's Kappa ($\kappa$), Macro-F1 (MF1), Sensitivity (Sens.) and Specificity (Spec.). Notably, the scaling effect exhibits diminishing returns, suggesting that while larger model sizes capture increasingly complex physiological patterns inherent in sleep data, the incremental gains become smaller with each parameter increase. Overall, these results affirm the robustness and scalability of the `sleep2vec` architecture, indicating its suitability for capturing detailed, multimodal physiological dynamics in sleep studies.

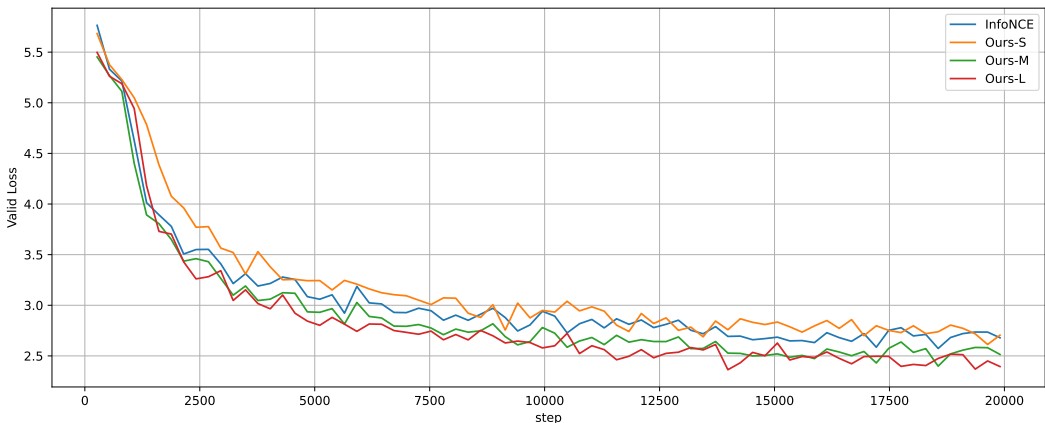

Figure 7: The validation loss curves compare a vanilla InfoNCE baseline with DASH-InfoNCE models at three scales, small, medium, and large, denoted as Ours-S, Ours-M, and Ours-L, respectively. As model scale increases, training consistently converges to lower validation loss. Moreover, at matched model scales, DASH-InfoNCE achieves lower validation loss than the comparable InfoNCE baseline.

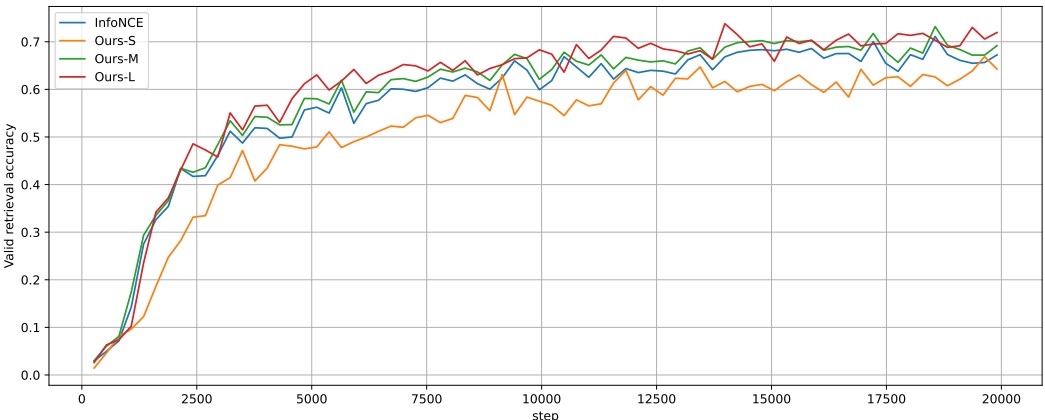

Figure 8: The validation retrieval accuracy curves compare a vanilla InfoNCE baseline with DASH-InfoNCE models at three scales, small, medium, and large, denoted as Ours-S, Ours-M, and Ours-L, respectively. Retrieval accuracy consistently improves as model scale increases. Moreover, at matched model scales, DASH-InfoNCE achieves higher retrieval accuracy than the corresponding InfoNCE baseline, demonstrating more effective cross-modal alignment.

The validation loss curves in Figure 7 and Figure 8 further support the scaling analysis above. Beyond the expected trend that larger models attain lower validation loss and higher retrieval accuracy, a key additional finding is that DASH-InfoNCE consistently yields improved optimization behavior and superior retrieval performance relative to the vanilla InfoNCE objective when evaluated at comparable model scales.

### A.5.2 SCALING LAW OF PRE-TRAINING DATA SIZE

The impact of pre-training data scale on cross-cohort generalization is examined by varying the fraction of the pre-training corpus while keeping the downstream fine-tuning protocol fixed. Models are pre-trained using 25%, 50%, 75% and 100% of the available data and subsequently evaluated on an unseen cohort.

As shown in Table 11 and Table 12, increasing the amount of pre-training data leads to consistent improvements in downstream performance, particularly in Macro-F1 (MF1) and Cohen's $\kappa$. The

Table 11: Effect of pre-training data size on sleep staging performance (W/N1/N2/N3/REM). All experimental setups, training protocols, and evaluation metrics follow those reported in Table 7, with retrieval accuracy ("Retrieval Acc.") computed on the same validation set. All medium-sized sleep2vec models are fine-tuned and evaluated on **SHHS**. **Bold numbers** denote the best performance among FMs.

| Data Fraction | Overall Performance (↑) | | | | | Class-wise F1 (↑) | | | | | Retrieval Acc. (↑) |
|---|---|---|---|---|---|---|---|---|---|---|---|
| | Acc. | $\kappa$ | MF1 | Sens. | Spec. | Wake | N1 | N2 | N3 | REM | |
| 25% | 87.1 | 0.82 | 77.2 | 76.3 | 96.4 | 93.7 | 37.6 | 86.8 | 77.7 | 89.9 | 0.281 |
| 50% | 88.0 | 0.83 | 77.9 | 76.8 | 96.6 | 94.3 | 38.5 | 87.5 | 78.8 | 90.6 | 0.295 |
| 75% | 88.4 | 0.83 | 78.7 | 77.1 | 96.7 | 94.6 | 41.2 | 88.0 | 78.3 | 91.1 | 0.340 |
| 100% | **88.6** | **0.84** | **79.5** | **78.4** | **96.8** | **94.8** | **44.1** | **88.2** | **79.2** | **91.2** | **0.368** |

Table 12: Effect of pre-training data size on cross-cohort sleep staging performance (W/N1/N2/N3/REM). All experimental setups, training protocols, and evaluation metrics follow those reported in Table 7, with retrieval accuracy ("Retrieval Acc.") computed on the same validation set. All medium-sized sleep2vec models are fine-tuned on **SHHS** and evaluated on the **unseen APPLES**. **Bold numbers** denote the best performance among FMs.

| Data Fraction | Overall Performance (↑) | | | | | Class-wise F1 (↑) | | | | | Retrieval Acc. (↑) |
|---|---|---|---|---|---|---|---|---|---|---|---|
| | Acc. | $\kappa$ | MF1 | Sens. | Spec. | Wake | N1 | N2 | N3 | REM | |
| 25% | 76.1 | 0.65 | 62.7 | 70.9 | 93.1 | 88.5 | 23.1 | 79.5 | 36.2 | 86.4 | 0.281 |
| 50% | 76.8 | 0.66 | 63.0 | 71.7 | 93.3 | 89.5 | 21.6 | 80.0 | 36.2 | 87.7 | 0.295 |
| 75% | 78.0 | 0.68 | 64.6 | 71.7 | 93.6 | 89.1 | 25.4 | 81.5 | 38.8 | 88.0 | 0.340 |
| 100% | **78.4** | **0.69** | **65.2** | **72.0** | **93.8** | **89.6** | **27.3** | **81.8** | **39.0** | **88.2** | **0.368** |

performance improvements are most substantial when moving from low to intermediate-scale data regimes, with diminishing returns as the full dataset is utilized. This trend suggests that larger-scale physiological pre-training promotes more robust and transferable representations, which is especially valuable under distribution shift.

Collectively, these findings indicate that the proposed framework benefits notably from increased data scale and exhibits stable generalization properties across cohorts.

### A.5.3 SCALING LAW OF PRE-TRAINING MODALITY NUMBER

Table 13: Effect of modality-scaling strategies on SHHS. "Single-stage" corresponds to the proposed medium sized sleep2vec model pre-trained from scratch using all available modalities. "Stage 1/2/3" implement a curriculum in which training begins with the most frequent and informative channels (EEG, RESP, and IBI in Stage 1), followed by the addition of EOG, ECG, and nasal airflow in Stage 2, and finally EMG, abdominal/thoracic belts, and SpO$_2$ in Stage 3, with each stage continuing from the previous checkpoint. All models are fine-tuned and evaluated on **SHHS** under identical EEG only and RESP+IBI downstream settings. **Bold numbers** indicate the best performance among FMs.

| PSG Channel Set | | Overall Performance (↑) | | | | | Class-wise F1 (↑) | | | |
|---|---|---|---|---|---|---|---|---|---|---|
| Inference Subset | Curriculum | Acc. | $\kappa$ | MF1 | Sens. | Spec. | Wake | N1 | N2 | N3 | REM |
| EEG | Stage 1 | 86.9 | 0.81 | 77.4 | 76.3 | 96.3 | 93.5 | 41.0 | 86.7 | 77.2 | 88.6 |
| | Stage 2 | 87.0 | **0.82** | **77.7** | 76.5 | 96.3 | 93.6 | **42.5** | 86.7 | 76.8 | 88.8 |
| | Stage 3 | 87.3 | **0.82** | 77.2 | 75.7 | 96.4 | 93.7 | 38.9 | **87.0** | 77.6 | **88.9** |
| | Single-stage | **87.4** | **0.82** | 77.3 | **76.6** | **96.5** | **94.2** | 40.1 | 86.5 | **77.7** | 88.3 |
| RESP+IBI | Stage 1 | 82.1 | 0.74 | 67.5 | 66.6 | 94.8 | 91.9 | 15.2 | 80.5 | 63.7 | 86.3 |
| | Stage 2 | 82.2 | **0.75** | 66.3 | 66.9 | 94.9 | 91.8 | 6.6 | 80.2 | **66.2** | 86.6 |
| | Stage 3 | 82.5 | **0.75** | **68.2** | **67.3** | 95.0 | **92.3** | **17.1** | 80.9 | 63.7 | **86.8** |
| | Single-stage | **83.0** | **0.75** | 65.9 | 65.8 | **95.1** | 86.6 | 5.3 | 80.9 | 64.3 | 86.6 |

Table 14: Effect of modality-scaling strategies evaluated on the unseen APPLES. All models are fine-tuned on **SHHS** and evaluated on **unseen APPLES** dataset under identical EEG only and RESP+IBI downstream settings. **Bold numbers** indicate the best performance among FMs.

| PSG Channel Set | | Overall Performance (↑) | | | | | Class-wise F1 (↑) | | | | |
|---|---|---|---|---|---|---|---|---|---|---|---|
| Inference Subset | Curriculum | Acc. | $\kappa$ | MF1 | Sens. | Spec. | Wake | N1 | N2 | N3 | REM |
| EEG | Stage 1 | 76.6 | 0.66 | 63.6 | **72.8** | 93.3 | 89.4 | 25.4 | 79.8 | 36.7 | 86.5 |
| | Stage 2 | **77.4** | **0.67** | **64.3** | 72.5 | **93.5** | **89.6** | **26.8** | 80.8 | 37.6 | 86.9 |
| | Stage 3 | 77.2 | **0.67** | 63.2 | 72.1 | 93.4 | 89.1 | 20.8 | 80.8 | **37.8** | **87.3** |
| | Single-stage | 76.7 | 0.66 | 62.5 | 71.9 | 93.3 | **89.6** | 19.1 | 80.0 | 37.3 | 86.7 |
| RESP+IBI | Stage 1 | 72.3 | 0.60 | 56.8 | 66.4 | 91.9 | 86.7 | 7.1 | 75.3 | 31.0 | 83.8 |
| | Stage 2 | 71.2 | 0.59 | 55.4 | 66.7 | 91.7 | 86.8 | 3.1 | 73.9 | 29.0 | 83.9 |
| | Stage 3 | 72.3 | 0.60 | 57.1 | **67.0** | 91.9 | **87.6** | 7.8 | 74.8 | 30.1 | **85.0** |
| | Single-stage | **73.2** | **0.61** | **57.8** | 66.2 | **92.1** | 86.5 | **10.2** | **76.5** | **31.5** | 84.2 |

The effect of pre-training modality count on downstream performance is examined by comparing four variants of the DASH-InfoNCE sleep2vec framework. The three curriculum stages differ only in the modality sets introduced during pre-training: **Stage 1** includes the most frequent and informative channels (EEG, RESP, and IBI); **Stage 2** resumes from the Stage 1 checkpoint and adds EOG, ECG, and nasal airflow; **Stage 3** incorporates the remaining, less frequent channels (EMG, abdominal/thoracic belts, and $SpO_2$). By contrast, the Single-stage model is trained from scratch using all modalities simultaneously.

All variants are fine-tuned using identical protocols on SHHS with either EEG or RESP+IBI as downstream inputs, and are evaluated both in-domain on SHHS and cross-cohort on APPLES. As reported in Table 13 and Table 14, expanding the modality set during pre-training yields small but consistent gains over the Stage 1 baseline, with the Single-stage model generally attaining the strongest or near-strongest performance. Importantly, the addition of rarer modalities in Stage 3 does not degrade performance, and all variants fall within a narrow accuracy and Macro-F1 (MF1) range. This stability indicates that sleep2vec scales predictably and robustly with respect to both modality number and modality diversity.

Moreover, as the number of pre-training modalities increases, the model retains, and often enhances, its performance when fine-tuned using only the originally available modalities, providing greater flexibility in downstream modality selection without sacrificing accuracy.

## A.6 RETRIEVAL ACCURACY MATRIX

The alignment quality is qualitatively validated using average recall@1 metrics computed across modalities on a test set comprising 10,109 segments. `sleep2vec` achieves a recall@1 of 36.8%, significantly surpassing the baseline with the original InfoNCE loss, which achieves 35.1%. This highlights the representational quality and discriminative capacity of the obtained embeddings.

The Recall@1 cross-modal retrieval accuracy matrix visualized in Figure 9 demonstrates distinct modality-specific alignment patterns. Notably, Respiratory and ABD/Thor exhibit exceptionally high mutual retrieval accuracy (0.82), aligning well with their known physiological coupling. EEG also demonstrates strong alignment with EOG (0.69). Conversely, modalities such as $SpO_2$ and EMG generally yield lower retrieval accuracies ($\approx$ 0.2–0.4), reflecting comparatively weaker physiological correlations.

## A.7 COMPLEMENTARY DOWNSTREAM FINE-TUNING RESULTS

### A.7.1 SLEEP STAGING CONFIGURATIONS

For fine-tuning the transformer backbone in the sleep staging task, we utilized Low-Rank Adaptation (LoRA) to achieve parameter-efficient adaptation. Specifically, LoRA adapters were integrated into the `query`, `key` and `value` projections of every transformer layer, while keeping the original backbone parameters frozen. Unless specified otherwise, we set the rank to $r = 8$, scaling factor $\alpha = 16$, dropout probability $p = 0.05$, without incorporating additional biases. For multimodal

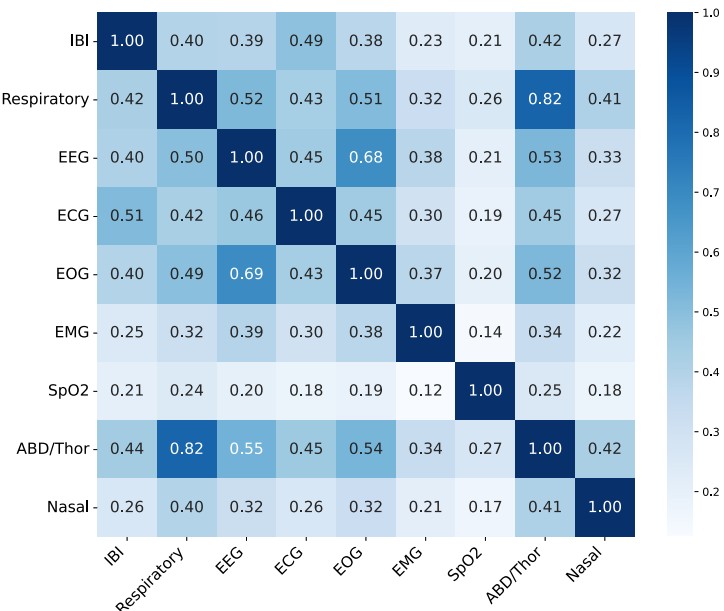

Figure 9: Recall@1 retrieval accuracy matrix of the learned representations. Rows correspond to the query modality, while columns indicate the retrieved modality.

fine-tuning scenarios, the same set of LoRA adapters was shared across all modalities. Instead of relying on a special classification token, transformer output embeddings from the final layer at each time step were individually projected through a two-layer MLP classifier, whose hidden dimension matched that of the backbone output. Optimization was performed using the AdamW optimizer with a learning rate of $1 \times 10^{-4}$ and a weight decay of $1 \times 10^{-5}$.

Baseline models are reproduced using their original hyperparameters as reported in the corresponding publications.

### A.7.2 PERFORMANCE ON WSC DATASET

Several trends emerge from Table 15:

**(i)** Comprehensive research on PSG data remains limited, as existing methods commonly focus on single-channel EEG or small subsets of physiological signals. Specialized models typically provide established benchmarks, particularly in cardiorespiratory modality configurations, presenting significant evaluation standards for generalized foundation models.

**(ii)** Foundation models consistently exhibit strong performance, often surpassing specialized sleep staging methods. This is evident in the EEG configuration, where sleep2vec notably achieves the best overall performance among foundation models (Accuracy: 86.3%, $\kappa$: 0.80), outperforming baseline FM SleepFM (Accuracy: 84.3%, $\kappa$: 0.76).

**(iii)** sleep2vec consistently demonstrates superior performance across various PSG channel subsets. Specifically, in the "IBI & RESP" channel configuration, sleep2vec substantially surpasses both specialized and baseline foundation models (Accuracy: 81.6% compared to SleepFounder's 79.8% and SleepFM's 77.7%). Similarly, in the "EEG & EOG & EMG" subset, sleep2vec outperforms baseline foundation models (Accuracy: 86.8% vs. 84.5%) and considerably surpasses specialized methods (Accuracy: 86.8% *vs.* 77.6%). In the Full Channels configuration, sleep2vec achieves the highest performance across multiple metrics (Accuracy: 87.3%, $\kappa$ :0.81), underscoring the effectiveness of leveraging comprehensive modality combinations.

To further assess cross-cohort generalization, we evaluate models fine-tuned on WSC directly on the APPLES cohort, which is unseen during both pre-training and fine-tuning. A similar trend is observed in Table 16, mirroring the cross-cohort generalization results reported in Table 2.

Table 15: Performance of five-class sleep staging (W/N1/N2/N3/REM) across PSG channel sets and models on **WSC**. Reported metrics regarding overall performance including Accuracy (Acc., %), Cohen Kappa ($\kappa$), Macro-F1 (MF1, %), Sensitivity (Sens., %) and Specificity (Spec., %). Class-wise F1 (%) is also listed. Baselines reproduced by us for fair comparison are marked with †. Note that these foundation model baselines were individually pre-trained for each PSG channel subset, whereas sleep2vec was pre-trained only once across all modalities. "FULL CHANNELS" refers to the fixed channel configuration that each model is designed for and individually pre-trained on. Other naming conventions follow the one adopted in Table 1. Underlined numbers indicate the best overall performance within each channel set; **bold numbers** denote the best performance among FMs; **bold-underlined numbers** indicate cases where the FM surpasses specialized models.

| PSG Channel Set | | Overall Performance (↑) | | | | | Class-wise F1 (↑) | | | | |
| --- | --- | --- | --- | --- | --- | --- | --- | --- | --- | --- | --- |
| Inference Subset | Model | Acc. | $\kappa$ | MF1 | Sens. | Spec. | W | N1 | N2 | N3 | REM |
| EEG | *Foundation Model* | | | | | | | | | | |
| | SleepFM (Thapa et al., 2024; 2025) † | 84.3 | 0.76 | 73.6 | 72.4 | 95.2 | 90.1 | 40.4 | **89.4** | **62.5** | 85.4 |
| | sleep2vec | **86.3** | **0.80** | **74.8** | **73.5** | **96.1** | **93.8** | **45.2** | 89.1 | 60.1 | **85.7** |
| IBI & RESP | *Specialized (non-FM) Model* | | | | | | | | | | |
| | Sun et al. (2019) † | 74.7 | 0.59 | 56.6 | 55.9 | 91.5 | 79.6 | 15.4 | 81.1 | 26.6 | 80.4 |
| | Goldammer et al. (2022) | 73.0 | 0.57 | 52.0 | 52.7 | 91.0 | 74.3 | 13.9 | 80.5 | 14.8 | 75.7 |
| | *Foundation Model* | | | | | | | | | | |
| | SleepFM (Thapa et al., 2024; 2025) † | 77.7 | 0.65 | 56.7 | 57.0 | 92.5 | 83.8 | 10.6 | 83.9 | 24.5 | 81.0 |
| | SleepFounder (Nie et al., 2025) † | 79.8 | 0.69 | 65.5 | 64.8 | 93.7 | 86.8 | 28.0 | **85.2** | 43.5 | **84.2** |
| | sleep2vec | **81.6** | **0.72** | **66.4** | **65.1** | **94.6** | **91.6** | **29.1** | 84.0 | **44.4** | 82.9 |
| EEG & EOG & EMG | *Specialized (non-FM) Model* | | | | | | | | | | |
| | Olesen et al. (2021) | 77.6 | 0.66 | — | — | — | — | — | — | — | — |
| | *Foundation Model* | | | | | | | | | | |
| | SleepFM (Thapa et al., 2024; 2025) † | 84.5 | 0.77 | 75.1 | 74.9 | 95.4 | 90.0 | 44.3 | 89.3 | 63.8 | 87.9 |
| | sleep2vec | **86.8** | **0.80** | **77.4** | **75.5** | **96.1** | **92.8** | **51.2** | **90.1** | **63.9** | **88.8** |
| FULL CHANNELS | *Foundation Model* | | | | | | | | | | |
| | SleepFM (Thapa et al., 2024; 2025) † | 84.6 | 0.77 | 75.4 | 75.0 | 95.4 | 90.0 | 45.2 | 89.4 | 64.1 | 88.1 |
| | PFTSleep (Fox et al., 2025) | 85.5 | 0.78 | 73.8 | 74.4 | 95.5 | 90.0 | 33.8 | 90.1 | 65.2 | 89.7 |
| | sleep2vec (InfoNCE) | 87.1 | 0.81 | 78.2 | 76.7 | 96.2 | 93.1 | 50.7 | 90.5 | 67.3 | 89.5 |
| | sleep2vec | **87.3** | **0.81** | **79.0** | **77.5** | **96.3** | **93.3** | **53.8** | **90.7** | **67.5** | **89.7** |

Table 16: Cross-cohort evaluation of five-class sleep staging (W/N1/N2/N3/REM) across PSG channel sets and models on **unseen APPLES**. Models are fine-tuned on WSC without seeing any data from APPLES during both pre-training and fine-tuning. "FULL CHANNELS" refers to the fixed channel configuration that each model is designed for and individually pre-trained on. Underlined numbers indicate the best overall performance within each channel set; **bold numbers** denote the best performance among FMs; **bold-underlined numbers** indicate cases where the FM surpasses specialized models.

| PSG Channel Set | | Overall Performance (↑) | | | | | Class-wise F1 (↑) | | | | |
| --- | --- | --- | --- | --- | --- | --- | --- | --- | --- | --- | --- |
| Inference Subset | Model | Acc. | $\kappa$ | MF1 | Sens. | Spec. | W | N1 | N2 | N3 | REM |
| IBI & RESP | *Specialized (non-FM) Model* | | | | | | | | | | |
| | (Sun et al., 2019) † | 69.9 | 0.54 | 51.6 | 53.3 | 90.4 | 78.6 | 8.9 | 76.2 | 17.1 | 77.2 |
| | (Goldammer et al., 2022) † | 68.6 | 0.52 | 48.6 | 50.3 | 90.2 | 74.3 | 8.1 | 76.7 | 14.6 | 69.5 |
| | *Foundation Model* | | | | | | | | | | |
| | SleepFM (Thapa et al., 2024; 2025) † | 74.4 | 0.61 | 56.3 | 56.5 | 91.9 | 81.8 | 17.0 | 81.1 | 20.2 | 81.3 |
| | SleepFounder (Nie et al., 2025) † | 74.0 | 0.61 | 59.4 | 64.0 | 92.1 | 84.7 | 17.9 | 79.2 | **31.7** | 83.5 |
| | sleep2vec (InfoNCE) | 76.1 | 0.64 | 55.9 | 55.5 | 92.4 | 85.4 | 16.8 | 81.5 | 14.4 | 81.5 |
| | sleep2vec | **76.4** | **0.64** | **57.7** | **57.7** | **92.7** | **85.7** | **19.2** | **81.8** | 19.5 | **82.5** |
| FULL CHANNELS | *Foundation Model* | | | | | | | | | | |
| | SleepFM (Thapa et al., 2024; 2025) † | 78.7 | 0.68 | 63.5 | 62.1 | 93.4 | 86.0 | 30.7 | **85.4** | 32.3 | 83.1 |
| | sleep2vec (InfoNCE) | 77.5 | 0.68 | 64.8 | **70.5** | 93.7 | 87.5 | 30.8 | 82.4 | 37.5 | **86.0** |
| | sleep2vec | **80.1** | **0.71** | **66.3** | 66.5 | **94.1** | **89.1** | **33.0** | 85.3 | **38.9** | 85.2 |

### A.7.3 INTERPRETABILITY OF CHANNEL-WISE CONTRIBUTION

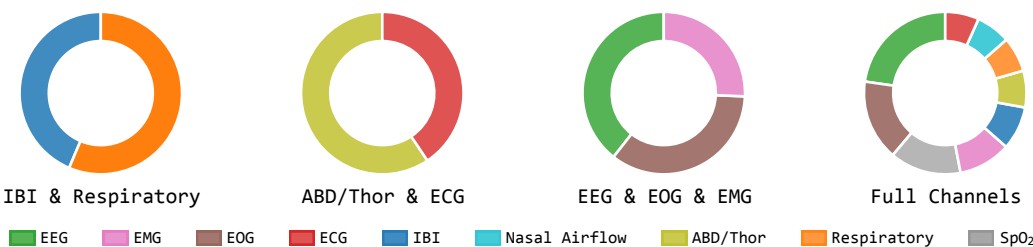

Figure 10: Visualization of modality-specific weights learned by the Gating Mechanism based feature fusion for the four SHHS sleep staging configurations in Table 1.

Beyond performance of sleep staging, the Gated scalar fusion used during fine-tuning provides a transparent, modality-level attribution of the downstream decision. Concretely, the learned scalars quantify each modality's contribution to the task-specific representation.

To illustrate, we analyze SHHS sleep staging under the four modality configurations in Table 1. Figure 10 presents the normalized fusion weights (visualized with a sharpening factor $T = 0.4$ that clarifies display while preserving relative ratios). Across tasks, EEG receives the largest weight, consistent with sleep staging being primarily annotated from EEG. EOG and EMG provide substantial complementary signal, while cardiorespiratory channels (e.g., airflow, ABD/Thor belt, IBI) carry smaller weights and $SpO_2$ contributes the least.

These weights are global, task-level attributions rather than per 30-second explanations, and they do not capture higher-order interactions between modalities. Nevertheless, these weights offer an interpretable summary of channel contribution that aligns with domain expectations and can inform sensor selection in channel-limited deployments.

### A.7.4 CLINICAL DIAGNOSIS CONFIGURATIONS

For clinical diagnosis tasks, the backbone parameters were kept frozen without applying LoRA adapters. Predictions were derived from the [CLS] token output of the final transformer layer, subsequently passed through a two-layer MLP classifier whose hidden dimension matched the backbone's output dimension. Training utilized the AdamW optimizer with a learning rate of $1 \times 10^{-4}$ and a weight decay of $1 \times 10^{-5}$, maintaining consistency with the sleep staging experimental setup. Note that site information is not used as a covariate during the fine-tuning process.

### A.8 LLM USAGE

We use LLMs to correct grammatical errors.

Some elements in Figure 1 (an illustration of a participant wearing a PSG device for sleep recording, along with icons representing each physiological signal modality) and Figure 2 (participant icons in bed) were generated using LLMs.

### A.9 ACKNOWLEDGMENT ON RESOURCE USAGE

The following acknowledgment information is listed as required by the National Sleep Research Resource (NSRR) (Zhang et al., 2018) and the individual datasets.

The Sleep Heart Health Study (SHHS) was supported by National Heart, Lung, and Blood Institute cooperative agreements U01HL53916 (University of California, Davis), U01HL53931 (New York University), U01HL53934 (University of Minnesota), U01HL53937 and U01HL64360 (Johns Hopkins University), U01HL53938 (University of Arizona), U01HL53940 (University of Washington), U01HL53941 (Boston University), and U01HL63463 (Case Western Reserve University). The National Sleep Research Resource was supported by the National Heart, Lung, and Blood Institute (R24 HL114473, 75N92019R002).

The National Heart, Lung, and Blood Institute provided funding for the ancillary MrOS Sleep Study, "Outcomes of Sleep Disorders in Older Men," under the following grant numbers: R01 HL071194, R01 HL070848, R01 HL070847, R01 HL070842, R01 HL070841, R01 HL070837, R01 HL070838, and R01 HL070839. The National Sleep Research Resource was supported by the National Heart, Lung, and Blood Institute (R24 HL114473, 75N92019R002).

The Multi-Ethnic Study of Atherosclerosis (MESA) Sleep Ancillary study was funded by NIH-NHLBI Association of Sleep Disorders with Cardiovascular Health Across Ethnic Groups (RO1 HL098433). MESA is supported by NHLBI funded contracts HHSN268201500003I, N01-HC-95159, N01-HC-95160, N01-HC-95161, N01-HC-95162, N01-HC-95163, N01-HC-95164, N01-HC-95165, N01-HC-95166, N01-HC-95167, N01-HC-95168 and N01-HC-95169 from the National Heart, Lung, and Blood Institute, and by cooperative agreements UL1-TR-000040, UL1-TR-001079, and UL1-TR-001420 funded by NCATS. The National Sleep Research Resource was supported by the National Heart, Lung, and Blood Institute (R24 HL114473, 75N92019R002).

This Wisconsin Sleep Cohort Study was supported by the U.S. National Institutes of Health, National Heart, Lung, and Blood Institute (R01HL62252), National Institute on Aging (R01AG036838, R01AG058680), and the National Center for Research Resources (1UL1RR025011). The National Sleep Research Resource was supported by the U.S. National Institutes of Health, National Heart Lung and Blood Institute (R24 HL114473, 75N92019R002).

The Apnea Positive Pressure Long-term Efficacy Study (APPLES) was supported by the National Heart, Lung, and Blood Institute (U01HL68060). The National Sleep Research Resource was supported by the U.S. National Institutes of Health, National Heart Lung and Blood Institute (R24 HL114473, 75N92019R002).

