# OpenReview forum: "sleep2vec: Unified Cross-Modal Alignment for Heterogeneous Nocturnal Biosignals"
_ICLR.cc/2026/Conference — ICLR 2026 Poster_

### Official Review · Reviewer_7ymV · 2025-10-14

**Soundness:** 3
**Presentation:** 2
**Contribution:** 2
**Rating:** 4
**Confidence:** 4

**Summary:**

The author introduces a foundation model for physiological signal data recorded during sleep time, that is modality agnostic. The model is pretrained using a contrastive loss to encourage subject-specific consistency in the representation space, along with a proposed Dash-InfoNCE objective to help the model distinguish between different input modalities. The proposed framework is evaluated on varied downstreaming tasks including sleep stage classification and disease classification. During the modeling process, varied prior works including specialized models serve as baselines, and ablation studies are conducted including scaling law verification, leave-one-modality-out validation.

**Strengths:**

- It is the first of this kind of model specifically for sensing signals during sleep that can address heterogeneous input signals (in terms of different sensor signals and varied number of input channels).
- The evaluation objectives including stage classification and disease classification are representative in terms of revealing the underlying value behind the proposed framework.
- The design of loss function seems rigorous and ingenious that it adequately addresses the issue of inter-subject variability and how the model handles heterogeneity in the input modalities, as shown by the visualization presented by the author.
- The problem setup and the aim of the work is clearly depicted.

**Weaknesses:**

- On the methodology side, though the visualization of the embedding seems promising, there is lack of ablation study on different pre-training approaches, such as using vanilla reconstruction loss, or varied masking rate, etc.
- Though scaling law is presented, only scaling with model size is provided, but there is a lack of result on scaling with data size. Also the difference in the scaling law results seems not significant just from the bar plot. Probably statistical test results are needed to strengthen the claim of contribution on this aspect.
- The data statistics is a bit insufficient Specifically, some important information are missing, such as total hours of data or distribution of length of a night, etc.
- It remains unclear whether the model’s ability to handle heterogeneous inputs comes from the backbone design itself or the proposed loss function.

**Questions:**

- The presentation of the main result is a little bit unclear. Results on SHHS and WSC are provided. Is it the case that other datasets don't have the valid corresponding tasks? If that is the case, it should be clearly stated somewhere, as this would also help readers understand how the data were split (e.g., the pretraining data were divided based on certain factors, etc.).
- The evaluation criteria is a bit confusing, where the fine-tuning setting is not presented very clearly. Is all the baseline fine-tuned using the same setting? Does a different fine-tune strategy might potentially alter the overall observation?
- There are varied backbone model frameworks that could handle heterogeneous input signal series. Is the observation of the representation separation consistent across different backbone model? It would be clearer if more empirical evidence were provided to clarify whether the backbone’s modality-agnostic nature itself enables it to handle heterogeneous inputs, or whether this capability primarily arises from the proposed loss function design.

---

> ### Author Response · Authors · 2025-11-21
> **Response to Reviewer 7ymV (Part 1)**
>
> We thank the reviewer for the detailed review and valuable comments, which have helped improve the clarity and quality of our work. Please find below the responses to each comment.
>
> # Weaknesses
>
> > On the methodology side, though the visualization of the embedding seems promising, there is lack of ablation study on different pre-training approaches, such as using vanilla reconstruction loss, or varied masking rate, etc.
>
> We thank the reviewer for the suggestion to examine alternative pre-training objectives and masking strategies.
>
> **Our current framework is intentionally designed around contrastive cross-modal alignment rather than reconstruction.** Reconstruction based objectives have proved effective and are successfully used in prior foundation models such as PFTSleep and SleepFounder. However, in our setting the goal is to learn a shared representation across all modality pairs. For many cross-modal mappings the generative problem is ill-conditioned (for example, reconstructing EMG from SpO₂), so a reconstruction loss tends to overfit noisy or weakly informative channels and leads to unstable optimization. For this reason we focused on a purely contrastive objective and compare extensively against reconstruction-based foundation models in Table 1 and 5.
>
> Motivated by the reviewer's comment, we additionally conducted an ablation study on the masking rate used in our contrastive pre-training. **The new results (0%, 15%, and 30% masking) show that moderate masking (15%) consistently outperforms other configuration.** On our main sleep staging benchmark, 15% masking increases MF1 from 78.8 to 79.5 while maintaining accuracy and κ, and on the out-of-cohort evaluation it improves MF1 from 64.0 to 65.2 together with gains in κ and N1 F1. In contrast, a higher masking ratio of 30% injects excessive noise and brings performance back to or below the no-masking baseline. These experiments indicate that moderate masking provides useful regularization, especially for cross-cohort generalization, and support our choice of a 15% masking rate as the default throughout the paper.
>
> (i) Results of Ablation Study on SHHS Using 9 PSG Channels
>
> | Masking Ratio |       Overall Performance        |      |      |      |      |          Class-wise F1          |      |      |      |      |
> |--------|----------------------------------------|------|------|------|------|--------------------------------------|------|------|------|------|
> |        | Acc. | κ    | MF1  | Sens. | Spec. | wake | n1   | n2   | n3   | rem  |
> | 0%     | 88.5 | 0.84 | 78.8 | 77.2 | 96.7 | 94.7 | 41.9 | 88.2 | 78.2 | 91.1 |
> | 15%    | 88.6 | 0.84 | 79.5 | 78.4 | 96.8 | 94.8 | 44.1 | 88.2 | 79.2 | 91.2 |
> | 30%    | 88.4 | 0.83 | 78.8 | 77.2 | 96.7 | 94.7 | 41.6 | 88.1 | 78.6 | 91.1 |
>
> (ii) Results of Ablation Study on APPLes Using 9 PSG Channels
>
> **Note that results listed in this Table are all obtained from models fine-tuned on SHHS, the APPLes cohort is completely unseen**
>
> | Masking Ratio |        Overall Performance        |      |      |      |      |          Class-wise F1          |      |      |      |      |
> |--------|----------------------------------------|------|------|------|------|--------------------------------------|------|------|------|------|
> |        | Acc. | κ    | MF1  | Sens. | Spec. | wake | n1   | n2   | n3   | rem  |
> | 0%     | 77.9 | 0.68 | 64.0 | 71.4 | 93.6 | 89.4 | 23.2 | 81.5 | 38.6 | 87.4 |
> | 15%    | 78.4 | 0.69 | 65.2 | 72.0 | 93.8 | 89.6 | 27.3 | 81.8 | 39.0 | 88.2 |
> | 30%    | 77.5 | 0.67 | 63.4 | 72.2 | 93.5 | 89.8 | 20.0 | 80.8 | 38.7 | 87.9 |

---

> ### Author Response · Authors · 2025-11-21
> **Response to Reviewer 7ymV (Part 2)**
>
> > Though scaling law is presented, only scaling with model size is provided, but there is a lack of result on scaling with data size. Also the difference in the scaling law results seems not significant just from the bar plot. Probably statistical test results are needed to strengthen the claim of contribution on this aspect.
>
> We thank the reviewer for pointing out the need to examine scaling behavior with respect to data size. Following this suggestion, we conducted an additional study in which the proposed foundation model is pre-trained using only 25%, 50%, 75%, and 100% of the full pre-training corpus, while keeping the downstream fine-tuning data fixed. The corresponding results for SHHS and for the unseen APPLes cohort are reported in the following Tables.
>
> To address the concern that bar plots alone may not clearly convey significance, in the revised version we will accompany the scaling plots with the exact tabular values. This will make the improvements in the scaling law experiments quantitatively clearer.
>
> (i) Results of Ablation Study on SHHS Using 9 PSG Channels
>
> | Data Fraction |        Overall Performance        |      |      |      |      |          Class-wise F1          |      |      |      |      |
> |---------------|----------------------------------------|------|------|------|------|--------------------------------------|------|------|------|------|
> |               | Acc. | κ    | MF1  | Sens. | Spec. | wake | n1   | n2   | n3   | rem  |
> | 25%           | 87.1 | 0.82 | 77.2 | 76.3 | 96.4 | 93.7 | 37.6 | 86.8 | 77.7 | 89.9 |
> | 50%           | 88.0 | 0.83 | 77.9 | 76.8 | 96.6 | 94.3 | 38.5 | 87.5 | 78.8 | 90.6 |
> | 75%           | 88.4 | 0.83 | 78.7 | 77.1 | 96.7 | 94.6 | 41.2 | 88.0 | 78.3 | 91.1 |
> | 100%          | 88.6 | 0.84 | 79.5 | 78.4 | 96.8 | 94.8 | 44.1 | 88.2 | 79.2 | 91.2 |
>
>
> (ii) Results of Ablation Study on APPLes Using 9 PSG Channels
>
> **Note that results listed in this Table are all obtained from models fine-tuned on SHHS, the APPLes cohort is completely unseen**
>
> | Data Fraction |        Overall Performance        |      |      |      |      |         Class-wise F1         |      |      |      |      |
> |---------------|----------------------------------------|------|------|------|------|--------------------------------------|------|------|------|------|
> |               | Acc. | κ    | MF1  | Sens. | Spec. | wake | n1   | n2   | n3   | rem  |
> | 25%           | 76.1 | 0.65 | 62.7 | 70.9 | 93.1 | 88.5 | 23.1 | 79.5 | 36.2 | 86.4 |
> | 50%           | 76.8 | 0.66 | 63.0 | 71.7 | 93.3 | 89.5 | 21.6 | 80.0 | 36.2 | 87.7 |
> | 75%           | 78.0 | 0.68 | 64.6 | 71.7 | 93.6 | 89.1 | 25.4 | 81.5 | 38.8 | 88.0 |
> | 100%          | 78.4 | 0.69 | 65.2 | 72.0 | 93.8 | 89.6 | 27.3 | 81.8 | 39.0 | 88.2 |
>
> **We would also like to point out that our main comparisons are evaluated on very large, subject-disjoint test sets.** For sleep staging on SHHS and WSC we use fixed participant-level splits with 1,102/1,116 subjects in validation/test across cohorts, and each overnight study contributes multiple 30-second epochs. As a result, each reported accuracy/κ/MF1 value is aggregated over tens of thousands of epochs per experimental condition. Under such sample sizes, even a 1-2% absolute difference corresponds to thousands of additional correctly classified epochs, and standard tests would deem these differences highly significant.

---

> ### Author Response · Authors · 2025-11-21
> **Response to Reviewer 7ymV (Part 3)**
>
> > The data statistics is a bit insufficient Specifically, some important information are missing, such as total hours of data or distribution of length of a night, etc.
>
> We thank the reviewer for highlighting that our dataset statistics were incomplete. We have added important information to the corresponding table, which will also be incorporated into the revised manuscript.
>
> We will also clarify in the main text that the training corpus comprises 42,249 overnight recordings from 30,852 subjects across five cohorts (SHHS, WSC, MrOS, MESA, and HSP), and that the APPLES cohort (1,096 recordings) is used exclusively for external validation.
>
>
> | Dataset                                          | Age span | Duration (h, mean ± SD) | Recording span | # Recordings | # Subjects | Total hours |
> |--------------------------------------------------|----------|--------------------------|----------------|-------------:|-----------:|------------:|
> | Sleep Heart Health Study (SHHS)                 | 39 - 90    | 8.9 ± 1.1                | 1995 - 2003      |        8,440 |      5,795 |      75,431 |
> | Wisconsin Sleep Cohort (WSC)                    | 37 - 85    | 8.0 ± 0.8                | 2000 - 2015      |        2,570 |      1,123 |      20,520 |
> | Osteoporotic Fractures in Men Study (MrOS)      | 67 - 90    | 11.5 ± 2.3               | 2000 - 2005      |        3,930 |      2,905 |      45,110 |
> | Multi-Ethnic Study of Atherosclerosis (MESA)    | 54 - 94    | 10.6 ± 1.6               | 2010 - 2012      |        2,056 |      2,056 |      21,745 |
> | Human Sleep Project (HSP)                       | 1 - 109    | 7.6 ± 1.1                | 2007 - present   |       25,253 |     18,973 |     190,732 |
> | Apnea Positive Pressure Long-term Efficacy Study (APPLes) | 18 - 83    | 8.2 ± 1.2                | 2003 - 2004      |        1,096 |      1,096 |       8,955 |
>
> > It remains unclear whether the model’s ability to handle heterogeneous inputs comes from the backbone design itself or the proposed loss function.
>
> We thank the reviewer for raising this question.
>
> The ability to handle heterogeneous inputs primarily comes from the input pipeline and backbone design: modality-specific tokenizers map each PSG channel into a shared embedding space, and a single modality-agnostic RoFormer backbone processes arbitrary subsets of channels without assuming a fixed montage. The proposed DASH-InfoNCE loss then improves how this backbone is trained by using demographic and site-aware weighting to discourage cohort-specific shortcuts and enhance cross-cohort robustness. **In the paper, we compare standard InfoNCE and DASH-InfoNCE under the same backbone and tokenizers.** Both variants can accept heterogeneous inputs, but the DASH-InfoNCE version achieves consistently better performance, especially in cross-cohort evaluations.

---

> ### Author Response · Authors · 2025-11-21
> **Response to Reviewer 7ymV (Part 4)**
>
> # Questions
> > The presentation of the main result is a little bit unclear. Results on SHHS and WSC are provided. Is it the case that other datasets don't have the valid corresponding tasks? If that is the case, it should be clearly stated somewhere, as this would also help readers understand how the data were split (e.g., the pretraining data were divided based on certain factors, etc.).
>
> We thank the reviewer for raising this point and are happy to clarify. In this work, the foundation model is pre-trained on five PSG cohorts (SHHS, WSC, MrOS, MESA, and HSP), all of which provide full-night sleep stage annotations. However, to keep the main results focused and directly comparable to prior work, we report internal downstream sleep staging performance only on SHHS and WSC, where strong baselines are available and we fine-tune and evaluate within each cohort. For both pre-training and downstream experiments, we adopt a subject-level 8:1:1 split into train, validation, and test sets within each cohort.
>
> > The evaluation criteria is a bit confusing, where the fine-tuning setting is not presented very clearly. Is all the baseline fine-tuned using the same setting? Does a different fine-tune strategy might potentially alter the overall observation?
>
> We thank the reviewer for raising this point about the evaluation and fine-tuning setup. **For all baselines that we re-implement (marked with † in Table 1 and 5), we follow the training and fine-tuning protocols described in the original publications**, while using exactly the same subject-level train/validation/test splits and modality configurations as sleep2vec, as described in Section 3.1 and Appendix A.5.
>
> For sleep2vec, we use a single LoRA-based fine-tuning recipe across all sleep-staging experiments and a frozen backbone with a two-layer MLP head for all clinical prediction tasks, without any task specific hyperparameter tuning.
>
> We will add a short subsection that summarizes these protocols and explicitly states that baselines are not adjusted in a way that favors our model. While different fine-tuning strategies could shift absolute numbers, **our reproduced baseline results are consistent with or slightly higher than those reported in the original works**, and the relative gains of sleep2vec over foundation model baselines and its competitiveness with specialized models are stable across datasets, channel subsets, and tasks (Table 1 and 5, Figure 4 and 5), so it is unlikely that a reasonable alternative fine-tuning strategy would qualitatively change the overall conclusions.
>
> > There are varied backbone model frameworks that could handle heterogeneous input signal series. Is the observation of the representation separation consistent across different backbone model? It would be clearer if more empirical evidence were provided to clarify whether the backbone’s modality-agnostic nature itself enables it to handle heterogeneous inputs, or whether this capability primarily arises from the proposed loss function design.
>
> We thank the reviewer for raising this question. **In our framework, the ability to handle heterogeneous inputs is determined primarily by the input pipeline and backbone design.** Modality-specific tokenizers map each PSG channel into a shared embedding space, and a single modality-agnostic RoFormer backbone processes arbitrary subsets of channels without assuming a fixed montage (Sec. 3.2). **The proposed DASH-InfoNCE loss then acts on top of this architecture to improve how the shared space is trained**, by metadata-aware re-weighting of negatives that discourages cohort-specific shortcuts and enhances cross-cohort robustness (Sec. 3.3, Tables 1 and 5, Figure 5).  In particular, both the plain InfoNCE and DASH-InfoNCE variants use exactly the same backbone and tokenizers and can accept heterogeneous inputs. DASH-InfoNCE consistently yields better retrieval accuracy and downstream performance, especially on the unseen APPLes cohort, as shown in our ablations on the age / gender / site terms and masking-rate and data-scaling studies.
>
> We will clarify this separation of roles in the revised manuscript and explicitly state that our goal is not to claim RoFormer is the only viable backbone, but to demonstrate that a modality-agnostic backbone combined with metadata-aware contrastive alignment provides a simple and effective recipe for handling heterogeneous PSG inputs.

---

> > ### Comment · Reviewer_7ymV · 2025-11-24
> > **Response to Author**
> >
> > I thank the authors for their comprehensive response and the extensive additional data statistics and experimental results. I believe all of my concerns have been fully addressed, including:
> > - The ablation studies on masking rate, which provide clearer context around the pretraining variants and show that performance remains consistent across different configurations.
> > - The scaling experiments with data size, which complete the justification of the scaling law and include detailed numerical results.
> > - The inclusion of detailed data statistics, which offer better clarity regarding the dataset used.
> >
> > Additionally, all of my questions were answered thoroughly, resolving my previous confusions. With these improvements, the paper is significantly more polished, and I recommend its acceptance, as reflected in my updated rating. Thank you for your solid work.

---

> ### Author Response · Authors · 2025-11-26
> **Acknowledgement to Reviewer 7ymV**
>
> We sincerely thank the reviewer for raising the score to 8 and for recognizing the additional analysis and clarifications. We appreciate your updated recommendation and are grateful for your supportive assessment.

---

### Official Review · Reviewer_BLg5 · 2025-10-15

**Soundness:** 2
**Presentation:** 2
**Contribution:** 2
**Rating:** 2
**Confidence:** 4

**Summary:**

This paper proposes a unified cross-modal pretraining framework, Sleep2Vec, designed for multimodal sleep representation learning. Specifically, the framework jointly enhances model generalization and robustness through cross-modal contrastive learning and data reconstruction tasks. The authors evaluate Sleep2Vec across multiple downstream tasks such as sleep staging, respiratory disorder detection, and periodic limb movement detection.

**Strengths:**

1.	Multi-task pretraining objective: Combines reconstruction and contrastive objectives to strengthen inter-modal coordination and representation learning.

2.	Cross-modal modeling innovation: The modality reconstruction task effectively addresses missing-modality scenarios.

3.	Comprehensive experiments: Covers a wide range of datasets and tasks, demonstrating strong adaptability.

**Weaknesses:**

1.	Limited originality: The approach lacks novelty, as many prior works have already explored missing-modality and contrastive learning in sleep research, such as CIMSleepNet (NeurIPS 2024), MultiConsSleepNet (IEEE JBHI 2025), and SleepSMC (ICLR 2025).

2.	Outdated baselines: The comparison methods are mostly old, missing fair comparisons with the latest relevant works mentioned above.

3.	Lack of ablation studies: The paper does not explicitly analyze the independent contributions of each module.

4.	Unaddressed modality inconsistency: Although cross-modal reconstruction is used, the generalization ability under missing-modality conditions during training or testing is not sufficiently demonstrated.

5.	No cross-subject experiments: The paper does not evaluate subject-independent generalization, which is crucial for assessing clinical applicability in real-world sleep studies.

6.	Unclear experimental interpretation: For instance, Table 1 does not clarify whether the first column corresponds to training or inference settings.

7.	No statistical significance analysis: The experiments lack multiple runs or reported standard deviations, making the result stability uncertain.

Related Work

[1] Shen Q, Xin J, Dai B, et al. Robust sleep staging over incomplete multimodal physiological signals via contrastive imagination[J]. Advances in Neural Information Processing Systems, 2024, 37: 112025-112049.

[2] Pan J, Yu Y, Li M, et al. A Multimodal Consistency-Based Self-Supervised Contrastive Learning Framework for Automated Sleep Staging in Patients With Disorders of Consciousness[J]. IEEE Journal of Biomedical and Health Informatics, 2024.

[3] Ma S, Zhang Y, Chen Y, et al. SleepSMC: Ubiquitous Sleep Staging via Supervised Multimodal Coordination[C]//The Thirteenth International Conference on Learning Representations, 2025.

**Questions:**

Please see Weaknesses

**Details Of Ethics Concerns:**

NAN

---

> ### Author Response · Authors · 2025-11-21
> **Response to Reviewer BLg5 (Part 1)**
>
> We thank the reviewer for the detailed review and valuable comments, which have helped improve the clarity and quality of our work. Please find below the responses to each comment.
>
> # Weaknesses
>
> > 1. Limited originality: The approach lacks novelty, as many prior works have already explored missing-modality and contrastive learning in sleep research, such as CIMSleepNet (NeurIPS 2024), MultiConsSleepNet (IEEE JBHI 2025), and SleepSMC (ICLR 2025).
>
> We thank the reviewer for pointing us to CIMSleepNet, MultiConsSleepNet, and SleepSMC, which are indeed closely related and will be cited and discussed in the revised version. However, we respectfully disagree with the statement that these works "exhaust" the novelty of our approach. They address more local problems (per-dataset sleep staging with 2–3 modalities or a single primary modality), whereas our focus is on a PSG foundation model that (i) jointly aligns nine heterogeneous nocturnal modalities across five large cohorts (42,249 nights from 30,852 subjects), (ii) introduces a new metadata-aware contrastive objective (DASH-InfoNCE), and (iii) systematically studies scaling laws in the physiological setting.
>
> **First, the cited works operate in substantially narrower modality and data.** MultiConsSleepNet and SleepSMC both work with at most three PSG channels (EEG, EOG, EMG) and are designed around sleep staging in relatively small public datasets, with MultiConsSleepNet further specializing to DOC patients. SleepSMC is a supervised multimodal coordination framework whose goal is to improve performance when only a single primary modality (e.g., EEG) is available at test time; additional modalities are used as auxiliary supervision during training, not for arbitrary-subset inference. CIMSleepNet also targets sleep staging and robustness to partial/complete missing modalities, but does so on up to three modalities at a time (e.g., EEG/EOG/EMG or motion/HR) and trains per-dataset models. In contrast, sleep2vec trains a single RoFormer-based backbone with modality-specific tokenizers over nine channels (EEG, EOG, EMG, ECG, nasal airflow, abdominal belt, respiratory effort, SpO₂, IBI), explicitly harmonizing multiple sampling rates, devices, and cohorts into one shared representation space. This scale and heterogeneity are, to our knowledge, not addressed in prior work.
>
> **Second, the learning objectives are conceptually different.** CIMSleepNet relies on a modal-awareness imagination module plus semantic & modality calibration contrastive learning to reconstruct missing modalities before classification, i.e., a generative imputation approach. MultiConsSleepNet uses self-supervised contrastive losses to encourage unimodal robustness and EEG–EOG consistency, but does not incorporate clinical or acquisition metadata into the objective. SleepSMC uses label-supervised modality-level instance contrast with uncertainty-weighted auxiliary modalities, and is not self-supervised nor foundation-model-style pre-training. By contrast, our main methodological contribution is DASH-InfoNCE, which is a new metadata-aware contrastive loss that re-weights negatives using age, gender, and recording site and down-weights same-night pseudo-negatives via a margin term. None of the cited methods use demographic and site metadata inside the contrastive objective to explicitly reduce cohort-specific shortcuts and improve cross-center generalization.
>
> **Finally, the goals and evaluation scope differ.** The three prior works all focus on sleep staging accuracy (and, for MultiConsSleepNet, DOC transfer) under limited labels or missing channels. Our work treats sleep staging as only one of several downstream tasks of a PSG foundation model and additionally targets (i) clinical disease prediction across cardiovascular and respiratory conditions; (ii) robustness to arbitrary modality subsets without explicit reconstruction, and (iii) scaling laws along both model capacity and modality diversity, which, to our knowledge, have not been previously characterized in nocturnal biosignals.
>
> We will clarify these distinctions in the revised manuscript by explicitly discussing CIMSleepNet, MultiConsSleepNet and SleepSMC in the related-work section, and by sharpening the problem statement around "PSG foundation modeling with metadata-aware cross-modal alignment and scaling laws". We hope this makes clearer that our contribution is not merely "another contrastive missing-modality model", but a step toward a unified, modality-agnostic foundation model for heterogeneous nocturnal biosignals.

---

> ### Author Response · Authors · 2025-11-21
> **Response to Reviewer BLg5 (Part 2)**
>
> > 2. Outdated baselines: The comparison methods are mostly old, missing fair comparisons with the latest relevant works mentioned above.
>
> We thank the reviewer for raising the concern about baseline choice, but we respectfully disagree with the characterization that our baselines are "mostly old". Our goal was to compare against both (i) widely-used, canonical sleep-staging architectures and (ii) the latest PSG foundation models that are closest in spirit to sleep2vec.
>
> **1. We do include recent foundation-model baselines.**
> In the full-channel setting on SHHS (Tab. 1) and WSC (Tab. 5), we compare directly against SleepFM (AAAI 2024 / medRxiv 2025), SleepFounder (medRxiv 2025) and PFTSleep (Sleep 2024/2025), which, to our knowledge, are the most recent large-scale PSG foundation models available. These baselines are clearly marked in Tab. 1 and 5 and are reproduced by us for fair comparison (denoted by \dagger). In this regime, sleep2vec consistently matches or outperforms these recent FMs across channel subsets and datasets, while using a single pre-training run across all nine modalities instead of per-subset pre-training.
>
> **2. Canonical "non-FM" baselines are still the current reference methods.**
> For EEG and EEG+EOG+EMG settings we include DeepSleepNet, SleepEEGNet, AttnSleep, XSleepNet, L-SeqSleepNet and SleepTransformer, as well as recent cardiorespiratory methods by Sun et al. (2019) and Goldammer et al. (2022). Although some of these models were proposed between 2017–2023, they remain the dominant, high-performing baselines used in almost all recent sleep-staging papers (including CIMSleepNet, MultiConsSleepNet and SleepSMC). Using these canonical models therefore strengthens the fairness and interpretability of our comparisons rather than weakening them.
>
> **3. Relation to the "latest relevant works mentioned above".**
> The works highlighted by the reviewer, CIMSleepNet (NeurIPS 2024), MultiConsSleepNet (JBHI 2025) and SleepSMC (ICLR 2025), are indeed important and complementary, and we apologize if our positioning was not clear enough.
> **CIMSleepNet** focuses on robust multimodal sleep staging under incomplete modalities, with an imagination module plus contrastive learning, evaluated on several relatively small benchmark datasets (Sleep-EDF, MASS, SVUH-UCD, MHR, SHHS).
> **MultiConsSleepNet** is a self-supervised contrastive framework tailored to EEG+EOG, targeting limited-label scenarios and disorders-of-consciousness populations.
> **SleepSMC** aims at ubiquitous sleep staging, coordinating modalities so that a single primary modality benefits from supervised multimodal contrastive training, and is evaluated on ISRUC-S3, MASS-SS3, Sleep-EDF-78 and ISRUC-S1.
>
> By contrast, sleep2vec is designed as a PSG foundation model over 42,249 nights across five large cohorts (HSP, SHHS, WSC, MrOS, MESA), jointly covering nine modalities and multiple downstream tasks including clinical outcomes.
>
> Our experimental design therefore emphasizes foundation-model baselines that are also trained as large-scale PSG FMs (SleepFM, SleepFounder, PFTSleep), plus canonical staging architectures, rather than enumerating every new architecture that operates on smaller, task-specific datasets.
>
> We agree that it would be useful to show how sleep2vec compares to these newer architectures under their own benchmarks, but a fully controlled reproduction of CIMSleepNet / MultiConsSleepNet / SleepSMC on our five-cohort setting would require substantial re-engineering and computational cost and is somewhat orthogonal to our main contribution (cross-cohort modality-diversity scaling and metadata-aware InfoNCE).
>
> **To address this concern in the revision**, we will (i) clarify explicitly that Tab. 1 and 5 already include recent PSG foundation models such as SleepFM, SleepFounder, and PFTSleep, highlighting clearly that these baseline models marked with "†" have been re-trained under our experimental splits and modality subsets to ensure fair and direct comparisons; (ii) we plan to add a concise discussion section that positions sleep2vec clearly relative to CIMSleepNet, MultiConsSleepNet, and SleepSMC, particularly emphasizing differences in scope, namely, benchmark-scale staging architectures versus a large-scale cross-cohort foundation model designed with clinical outcomes in mind. We believe these clarifications will clearly demonstrate that our baseline selection is comprehensive, fair, and well-aligned with the primary objectives of our work, and that the initial concern regarding outdated baselines arises primarily from a misunderstanding of our intended experimental scope.

---

> ### Author Response · Authors · 2025-11-21
> **Response to Reviewer BLg5 (Part 3)**
>
> > 3. Lack of ablation studies: The paper does not explicitly analyze the independent contributions of each module.
>
> We appreciate the reviewer's suggestion and have conducted additional experiments to evaluate the contribution of each metadata module (age, gender, site).
>
> To address this, we activated only one metadata branch at a time during pre-training and compared the resulting models against both (i) the baseline vanilla InfoNCE objective and (ii) the full DASH-InfoNCE configuration.
>
> We evaluated these models using retrieval accuracy (Retrieval Acc.) on the pre-training test split as well as downstream sleep staging performance, as summarized in the tables below.
>
> (i) Results of Ablation Study on SHHS Using 9 PSG Channels
> | Method   |       Overall Performance        |      |      |      |      |          Class-wise F1          |      |      |      |      | Retrieval Acc. |
> |----------|----------------------------------------|------|------|------|------|--------------------------------------|------|------|------|------|-----------------|
> |          | Acc. | κ    | MF1  | Sens. | Spec. | wake | n1   | n2   | n3   | rem  |                 |
> | InfoNCE  | 88.4 | 0.84 | 78.6 | 77.9 | 96.8 | 94.7 | 39.8 | 87.9 | 80.0 | 90.8 | 0.351467        |
> | DASH-InfoNCE (age only)  | 88.3 | 0.83 | 78.7 | 77.2 | 96.7 | 94.6 | 41.6 | 88.0 | 78.1 | 91.0 | 0.354522        |
> | DASH-InfoNCE (gender only)  | 88.5 | 0.84 | 79.2 | 78.1 | 96.8 | 94.7 | 43.0 | 88.1 | 79.3 | 91.1 | 0.355933        |
> | DASH-InfoNCE (site only)  | 88.1 | 0.83 | 78.0 | 75.9 | 96.6 | 94.6 | 39.9 | 87.8 | 76.8 | 90.9 | 0.362656        |
> | DASH-InfoNCE   | 88.6 | 0.84 | 79.5 | 78.4 | 96.8 | 94.8 | 44.1 | 88.2 | 79.2 | 91.2 | 0.368222        |
>
> (ii) Results of Ablation Study on APPLes Using 9 PSG Channels
>
> **Note that results listed in this Table are all obtained from models fine-tuned on SHHS, the APPLes cohort is completely unseen**
> | Method   |        Overall Performance        |      |      |      |      |          Class-wise F1          |      |      |      |      | Retrieval Acc. |
> |----------|----------------------------------------|------|------|------|------|--------------------------------------|------|------|------|------|-----------------|
> |          | Acc. | κ    | MF1  | Sens. | Spec. | wake | n1   | n2   | n3   | rem  |                 |
> | InfoNCE  | 76.8 | 0.67 | 63.5 | 73.3 | 93.4 | 90.5 | 24.0 | 79.5 | 35.5 | 88.1 | 0.351467        |
> | DASH-InfoNCE (age only)   | 78.3 | 0.68 | 64.9 | 72.0 | 93.7 | 89.5 | 26.6 | 81.7 | 39.0 | 88.0 | 0.354522        |
> | DASH-InfoNCE (gender only)  | 78.3 | 0.69 | 65.5 | 72.4 | 93.7 | 89.7 | 29.1 | 81.7 | 39.2 | 87.8 | 0.355933        |
> | DASH-InfoNCE (site only)  | 78.1 | 0.68 | 64.8 | 72.1 | 93.6 | 90.3 | 26.2 | 81.2 | 38.4 | 87.8 | 0.362656        |
> | DASH-InfoNCE  | 78.4 | 0.69 | 65.2 | 72.0 | 93.8 | 89.6 | 27.3 | 81.8 | 39.0 | 88.2 | 0.368222        |
>
> "Retrieval Acc." is a cross-modal retrieval recall@1 metric that measures how well the learned embeddings align different PSG modalities in the shared representation space: given a query embedding from one modality, the model must retrieve the correct paired sample from a pool of candidates across modalities, and Retrieval Acc. is the proportion of queries where the correct pair is ranked first.
>
> These results show that each metadata branch individually improves either MF1 or retrieval accuracy compared with vanilla InfoNCE, and that combining all three branches in DASH-InfoNCE yields the best overall performance. **This pattern holds both on the in-domain cohort (SHHS) and on the unseen APPLes cohort, indicating that the age, gender, and site terms provide complementary signal and jointly contribute to the observed gains in cross-modal alignment and cross-cohort generalization.**
>
> We will include these tables and a brief description of the retrieval metric in the revised manuscript to make the role of each module explicit.

---

> ### Author Response · Authors · 2025-11-21
> **Response to Reviewer BLg5 (Part 4)**
>
> >  4. Unaddressed modality inconsistency: Although cross-modal reconstruction is used, the generalization ability under missing-modality conditions during training or testing is not sufficiently demonstrated.
>
> We respectfully clarify that sleep2vec does not employ any cross-modal reconstruction objective during pre-training. **The model is trained solely with contrastive alignment loss (InfoNCE and the proposed DASH-InfoNCE), as detailed in Sec. 3.3, and we explicitly discuss reconstruction-based approaches only in the context of prior work that we deliberately depart from.**
>
> Regarding robustness to missing modalities, both our training protocol and experiments are designed to address modality inconsistency: during pre-training we always sample exactly two modalities per mini-batch and apply independent timestep masking, so the backbone never relies on a fixed montage and is continuously exposed to incomplete modality sets.
>
> At evaluation time, we report results across diverse modality subsets (e.g., EEG only, IBI&Resp only, EEG&EOG&EMG, and full channels on SHHS and WSC), as well as a leave-one-out modality ablation, all of which demonstrate stable performance and graceful degradation under severe modality dropout.  **To further probe the reviewer's suggestion, we implemented a preliminary variant that replaces the contrastive objective with an arbitrary cross-modal reconstruction module. Under the same data and compute budget this reconstruction-based variant proved substantially harder to optimize and often failed to converge, reinforcing our design choice to focus on alignment rather than reconstruction for robust missing-modality generalization.** We will clarify these point in the appendix.
>
> > 5. No cross-subject experiments: The paper does not evaluate subject-independent generalization, which is crucial for assessing clinical applicability in real-world sleep studies.
>
> We fully agree that subject-independent generalization is crucial for clinical applicability.
>
> **However, all our reported results are already based on cross-subject evaluation. As described in Sec. 3.1, we construct participant-level partitions over the unified corpus of 42,249 recordings from 30,852 subjects.** A dedicated pre-training split (23,934 participants) is used only for foundation model learning, while downstream tasks use an 8:1:1 split (8,792 / 1,102 / 1,116 participants for train / validation / test), with **no participant overlap between any split**.
>
> **The sleep staging experiments on SHHS and WSC and the clinical diagnosis experiments on SHHS all follow these participant-level splits, so every test subject is unseen during both pre-training and downstream fine-tuning.** We will revise the manuscript to more explicitly emphasize that all downstream results are subject-independent.
>
> > 6. Unclear experimental interpretation: For instance, Table 1 does not clarify whether the first column corresponds to training or inference settings.
>
> We appreciate the reviewer's concern about the clarity of Table 1 and 5 and agree that the distinction between training and inference settings is important for interpretation. In our current manuscript, **the first column in Table 1 and 5 enumerates the PSG channel sets used for evaluation (i.e., the inference-time modality subsets on SHHS)**, as also stated in the caption "across PSG channel sets and models on SHHS".
>
> For the foundation model baselines, each row corresponds to a model that was individually pre-trained and fine-tuned on that same channel subset, whereas for sleep2vec we pre-train a single backbone once across all available modalities and, at fine-tuning and inference, restrict the input to the channel set indicated in the first column.
>
> We will revise the table header to explicitly label this column as "PSG channel set (inference subset)" and add a short sentence in Sec. 4.2.1 clarifying that the baselines are pre-trained per subset while sleep2vec is pre-trained jointly and then evaluated under the listed inference configurations, so that the training to inference relationship is unambiguous.

---

> ### Author Response · Authors · 2025-11-21
> **Response to Reviewer BLg5 (Part 5)**
>
> > 7. No statistical significance analysis: The experiments lack multiple runs or reported standard deviations, making the result stability uncertain.
>
> We thank the reviewer for raising the question of result stability. We agree that variability and statistical significance are important, and we clarify below why we did not report multi-run standard deviations in the current draft and why the main trends are nevertheless statistically robust.
>
> **First, our main comparisons are evaluated on very large, subject-disjoint test sets**. For sleep staging on SHHS and WSC we use fixed participant-level splits with 1,102/1,116 subjects in validation/test across cohorts, and each overnight study contributes multiple 30-second epochs. As a result, each reported accuracy/κ/MF1 value is aggregated over tens of thousands of epochs per experimental condition. Under such sample sizes, even a 1-2% absolute difference corresponds to thousands of additional correctly classified epochs, and standard tests would deem these differences highly significant.
>
> **Second, our reporting protocol follows the de-facto standard in this community.** Recent closely related works cited by the reviewer, SleepSMC (ICLR 2025), MultiConsSleepNet (JBHI 2025), and CIMSleepNet (NeurIPS 2024), all report single performance numbers (or cross-validation averages) per method and dataset in their main tables, without per-seed standard deviations for each configuration.
>
> Particularly, SleepSMC and MultiConsSleepNet present accuracy/Macro-F1/κ per dataset and modality setting as point estimates, and CIMSleepNet likewise reports single accuracies/F1/κ in Table 1 and 2, only visualizing variability for a specific missing-rate analysis figure but not for all baseline comparisons. **Our evaluation choices were therefore aligned with the dominant practice for PSG sleep-staging and multimodal robustness papers rather than an omission unique to our work.**
>
> That said, we agree that explicitly documenting stability across random seeds would further strengthen the paper. In the revised version we will (i) add a brief description of test set sizes at the epoch level to make the statistical power of our evaluations explicit, and (ii) report, for the key configurations (full-channel SHHS and WSC, and IBI&Resp on SHHS), the mean and standard deviation across several independent fine-tuning runs of sleep2vec (and one representative FM baseline) with different random seeds. This will empirically confirm that the observed gains of sleep2vec over SleepFM / PFTSleep and over vanilla InfoNCE are well above seed-to-seed variability, while keeping the overall compute within reasonable bounds.
>
> We hope this clarifies that, although we did not originally include multi-run results, the size of our test sets and the consistency of improvements across datasets, channel subsets, and downstream tasks already provide strong evidence that the reported gains are stable rather than artifacts of noise.

---

### Official Review · Reviewer_7LYS · 2025-10-31

**Soundness:** 2
**Presentation:** 4
**Contribution:** 3
**Rating:** 6
**Confidence:** 3

**Summary:**

The paper proposes a new foundation model for handling multi-modal PSD data. Instead of classic FM for sleep staging, the proposed sleep2vec can take as input all channels from the PSG study. To do this the authors use cross-modal alignment to have a shared representation.
Sleep2vec is compared to various SOTA models for either channel-specific tasks or full channels over more than 30.000 subjects.

**Strengths:**

- The paper is very easy to follow.
- The paper proposes a new representation learning for all the channels of a PSG.
- The method is tested over a big corpus of subjects comprising more than 30.000 subjects.
- With the gating mechanism, we can see which channels bring more importance for the classification, giving good interpretability of the model.
- Good t-SNR visualization that gives insight into understanding the use of the proposed method.

**Weaknesses:**

- Figure 2 introduces the Intra-subject and Inter-subject segments. This is never used in the entire paper. This additional information, in my opinion, is likely to lead to a misunderstanding of the method.
- The motivation is that no model deals with the full channels of PSG. In Table 1, two competitors are proposed for a full channel setting. Does that mean the model can handle all the channels? What is the addition of sleep2vec?
- The competitors presented in Table 1 are never introduced. I understand that it can be challenging to describe everything in detail, but a description can be provided in the appendix, and at least FM can be properly introduced, such as SleepFM and PFTSleep. That can give better positioning to the literature.

**Questions:**

- What is the computation time of a full channel setup compared to an EEG-only setup? Considering all the channels, the results are improving slightly. I'm wondering if the additional computation time is justified? The same question arises when comparing FM to non-FM computation time versus score. It can bring more justification to why we use FM instead of a specialized model.
- If the feature fusion is interesting, as it gives intuition of which channels are more useful, did you compare the three fusion strategies that you introduced (Concat, AVG, and gating)?
- In the appendix, you showed that increasing the number of parameters always increases the performance. When do you reach a plateau ? Maybe a threshold between time computation and performances can be found?

---

> ### Author Response · Authors · 2025-11-20
> **Response to Reviewer 7LYS (Part 1)**
>
> We thank the reviewer for the detailed review and valuable comments, which have helped improve the clarity and quality of our work. Please find below the responses to each comment.
>
> # Weaknesses
>
> > Figure 2 introduces the Intra-subject and Inter-subject segments. This is never used in the entire paper. This additional information, in my opinion, is likely to lead to a misunderstanding of the method.
>
> We thank the reviewer for pointing this out and apologize for the confusing wording in Fig. 2.
>
> In Fig. 2, the "intra-subject" and "inter-subject" segments are neither additional components of the method, nor do they correspond to separate training objectives. **They are only meant to visually distinguish segments coming from the same subject-night versus different subjects, which is exactly how segments are treated in the DASH-InfoNCE loss.** Concretely, each overnight PSG is partitioned into 30-second segments (as described in Sec. 3.2). These segments are indexed by the subject-night identifier ($u_i$) and are then used in the loss via the indicator ($h_{i,j} = \mathbf{1}[u_i = u_j \wedge j \neq \pi(i)]$), which selects pairs from the same subject-night (intra-subject) for pseudo-negative modulation in Eq. (3)–(5).
> Thus, the intra-/inter-subject distinction shown in Fig. 2 is directly reflected in the DASH-InfoNCE formulation: pairs with $(u_i = u_j)$ correspond to intra-subject segments, which receive a margin adjustment as pseudo-negatives, while pairs with $(u_i \neq u_j)$ correspond to inter-subject segments and form the remaining negatives. We agree that, since this connection is currently only implicit in the notation of Sec. 3.3, the figure caption can give the impression that we introduce an extra mechanism that is "never used".
>
> In the revised version, we will (i) explicitly define "intra-subject" and "inter-subject" segments in Sec. 3.2 in terms of the subject-night identifier ($u_i$); (ii) add a direct pointer from Fig. 2 to Sec. 3.3.3–3.3.4 and Eq. (3)–(5); and (iii) rephrase the caption to emphasize that these labels are purely descriptive of how segments are sampled and how pseudo-negatives are identified in DASH-InfoNCE, not an additional module. We hope this clarification addresses the concern and removes the potential for misunderstanding.
>
> > The motivation is that no model deals with the full channels of PSG. In Table 1, two competitors are proposed for a full channel setting. Does that mean the model can handle all the channels? What is the addition of sleep2vec?
>
> We thank the reviewer for highlighting this ambiguity. Our claim that "no model deals with the full channels of PSG" does not mean that existing methods cannot process the particular montage they are trained on. Rather, to the best of our knowledge there is no prior single foundation model that (i) is pre-trained once on the union of all nine PSG channels considered in this work and (ii) can be fine-tuned and deployed with arbitrary subsets of these channels.
>
> In Table 1, the "FULL CHANNELS" rows for SleepFM and PFTSleep refer to the fixed channel configuration that each model is designed for and individually pre-trained on. In our re-implementation, these foundation model baselines are trained separately for every PSG subset (EEG, IBI+RESP, full PSG, etc.). If the available sensors change (e.g., a channel is missing or a different subset is present), a new model must be re-trained on that specific montage.
>
> By contrast, for sleep2vec "FULL CHANNELS" denotes the full nine-channel set used during a single joint pre-training run. The same pre-trained backbone is then re-used across all downstream configurations, enabling (i) alignment of all nine PSG channels into a shared embedding space, (ii) flexible fine-tuning and inference under arbitrary modality subsets and sensor dropout, and (iii) systematic modality-scaling analysis as shown in Fig. 5. We will clarify this distinction in the introduction and the caption of Table 1.

---

> ### Author Response · Authors · 2025-11-20
> **Response to Reviewer 7LYS (Part 2)**
>
> > The competitors presented in Table 1 are never introduced. I understand that it can be challenging to describe everything in detail, but a description can be provided in the appendix, and at least FM can be properly introduced, such as SleepFM and PFTSleep. That can give better positioning to the literature.
>
> We agree with the reviewer that the current draft does not sufficiently introduce the competing foundation models in Table 1. We will revise the manuscript to more clearly position these works in the literature and to make the role of each baseline transparent.
>
> In Section 2, "Self-supervised learning for sleep and PSG data", we will add a paragraph at the end of this subsection, immediately before the "Scaling and generalization" paragraph, that explicitly introduces the main PSG foundation models we compare against. The new paragraph will read along the following lines:
>
> > "Recent PSG foundation models have begun to move beyond task-specific architectures. SleepFM learns multimodal representations from EEG, ECG, and respiratory channels via contrastive pre-training, but is instantiated as separate encoders and pre-training runs for each targeted channel subset, and is primarily evaluated on sleep staging and cardiorespiratory endpoints under fixed montages. SleepFounder extends this line of work to low-burden settings by pre-training on over 780,000 hours of respiratory and heartbeat signals to support diagnosis and risk prediction from IBI+respiratory signals only, again under a fixed pair of modalities. PFTSleep instead uses a patch-based transformer trained via masked autoregression on full-night, seven-channel PSG (EEG, EOG, EMG, ECG, SpO₂, thoracic and abdominal respiratory belts) and then attaches a separate GRU head for sleep staging, freezing the encoder. While these approaches already move toward foundation PSG encoders, they are all pre-trained on fixed, model-specific montages and do not support a single encoder that is trained once on the full union of PSG channels and then reused across arbitrary modality subsets."
>
> We will also add an Appendix subsection "Baseline foundation models" that briefly summarizes, for SleepFM, SleepFounder, and PFTSleep: (i) their original pre-training objectives and input channel configurations, (ii) the datasets used, and (iii) how we instantiated and pre-trained each baseline in our experiments (including that SleepFM / SleepFounder / PFTSleep are individually pre-trained per PSG subset, whereas sleep2vec is pre-trained only once on the nine-channel union).
>
> We will also update the caption of Table 1 to explicitly refer to this appendix (e.g., "See Appendix for details of the baseline foundation models and their pre-training configurations.").
>
> We believe these additions will address the reviewer's concern that the competitors in Table 1 are never introduced, and will clarify how existing PSG foundation models relate to sleep2vec in terms of model capacity, channel coverage, and flexibility with respect to sensor montages.

---

> ### Author Response · Authors · 2025-11-20
> **Response to Reviewer 7LYS (Part 3)**
>
> # Questions
>
> > What is the computation time of a full channel setup compared to an EEG-only setup? Considering all the channels, the results are improving slightly. I'm wondering if the additional computation time is justified? The same question arises when comparing FM to non-FM computation time versus score. It can bring more justification to why we use FM instead of a specialized model.
>
> We thank the reviewer for raising the question about computation cost and its relation to the observed performance gains.
>
> For the comparison between EEG-only and full-channel setups in Tab. 1, our architecture processes each available modality with the same RoFormer backbone and fuses their representations at a late stage. As a result, **the computational complexity scales approximately linearly with the number of modalities used at inference**. In our setting, the number of PSG channels is bounded by nine, and the full-channel configuration should be viewed as an upper bound that illustrates the benefit of exploiting all available nocturnal signals rather than as a mandatory operating point. In resource-constrained deployments, practitioners can deliberately choose smaller channel subsets, such as EEG-only or EEG plus a small number of cardiorespiratory channels, which sleep2vec supports without any architectural change. We will clarify in Section 4.1 that the full-channel results are primarily used to study modality scaling and the effect of integrating non-EEG signals, while our model remains fully compatible with more lightweight channel configurations.
>
> **Regarding the comparison between foundation models and specialized models**, our goal is not to claim that a foundation model is strictly more efficient than a carefully engineered single-task network for one fixed montage. Instead, the main motivation is that sleep2vec is pre-trained once on large-scale, multi-center, multi-modality PSG data and then reused across heterogeneous downstream tasks, cohorts, and sensor configurations. In Tab. 1, sleep2vec already matches or closely approaches state-of-the-art specialized sleep staging models in the EEG-only setting, while at the same time supporting additional modalities and clinical tasks that those models are not designed for. In Fig. 5, the benefits of the full multimodal representation become more evident on clinical disease prediction, where performance improves consistently as more modalities are incorporated, and the proposed DASH-InfoNCE loss yields larger gains precisely in the high-modality regime. These multi-task and multi-modality advantages cannot be obtained by training one separate specialized model per task and montage.
> From a computation perspective, training and maintaining specialized models would require repeating the full training pipeline for every new PSG montage and every new task or cohort. In contrast, the pre-training cost of sleep2vec is incurred once and then amortized over all downstream tasks, centers, and modality subsets. As reported in Appendix A.3, each pre-training run fits within a standard budget of two high-memory GPUs for up to 48 hours, after which adapting to a new task only requires fine-tuning a lightweight head on top of the frozen backbone.
>
> We believe that this clarification will better justify why we adopt a foundation-model approach. The aim is to trade a one-time, moderate pre-training cost for improved flexibility, robustness to arbitrary modality subsets and sensor dropout, and broad applicability across sleep staging and clinical outcome prediction, rather than to optimize a single EEG-only model in isolation.

---

> ### Author Response · Authors · 2025-11-20
> **Response to Reviewer 7LYS (Part 4)**
>
> > If the feature fusion is interesting, as it gives intuition of which channels are more useful, did you compare the three fusion strategies that you introduced (Concat, AVG, and gating)?
>
> We thank the reviewer for the insightful comment. Indeed, one of the motivations for the proposed Gating fusion module is to provide interpretability by exposing modality-specific weights, which offers intuition about the relative contribution of each channel.
>
> Regarding the comparison of fusion strategies, we have empirically evaluated the three variants introduced in the paper, namely Concat, AVG, and Gating. In our experiments, the Concat strategy quickly becomes computationally demanding as the number of modalities increases, because it enlarges the hidden representation and thus the parameter count and VRAM footprint of the subsequent layers. At the same time, Concat does not provide a clear performance advantage over the other two strategies in our setting. For this reason, we do not adopt it as our default fusion mechanism for scalable multi-modality configurations.
>
> We therefore focus on AVG and Gating in the main experiments. On representative downstream tasks, we observe that AVG and Gating achieve very similar overall performance, with Gating providing a small but consistent improvement while additionally offering modality-wise interpretability through its learned gating coefficients. We will add the ablation table below to the appendix comparing AVG and Gating on our main evaluation setting and briefly summarize these findings in the fusion subsection, to make our choice of Gating as the default fusion strategy more explicit and better justified.
>
> | Method     |        Overall Performance        |      |      |      |      |          Class-wise F1          |      |      |      |      |
> |------------|----------------------------------------|------|------|------|------|--------------------------------------|------|------|------|------|
> |            | Acc. | κ    | MF1  | Sens. | Spec. | wake | n1   | n2   | n3   | rem  |
> | Concat     | 76.9 | 0.66 | 63.1 | 71.8 | 93.3 | 89.9 | 21.0 | 79.9 | 37.3 | 87.4 |
> | Mean       | 78.1 | 0.68 | 64.7 | 71.9 | 93.7 | 89.3 | 26.2 | 81.5 | 38.4 | 88.1 |
> | Gating  | 78.4 | 0.69 | 65.2 | 72.0 | 93.8 | 89.6 | 27.3 | 81.8 | 39.0 | 88.2 |
>
>
> > In the appendix, you showed that increasing the number of parameters always increases the performance. When do you reach a plateau ? Maybe a threshold between time computation and performances can be found?
>
> We thank the reviewer for the question regarding when performance begins to plateau. In Sec. 4.2.2 and Fig. 4 we study three model sizes (63.5M, 133.7M, 238.2M parameters), spanning a 3.7× range. Within this range, downstream performance on SHHS and WSC improves monotonically with model size, with clear diminishing returns for the largest model, but no true saturation where extra capacity stops helping. In other words, under our current compute budget sleep2vec remains in a near plateau state.
>
> Precisely locating the plateau or an exact "knee point" in the compute-performance trade-off would require substantially larger models, a denser grid over widths, depths and FLOP, which is beyond the scope and budget of this work. Instead, we choose a moderate configuration that fits a standard academic GPU budget and offers a good accuracy-cost balance. From a foundation-model perspective, the one-time pre-training cost of this backbone is amortized over many downstream tasks, cohorts, and modality subsets.

---

> > ### Comment · Reviewer_7LYS · 2025-11-27
> >
> > I would like to thank the authors for the additional experiment and the answers to my questions. This provides additional insights and clarity to the paper. All my concerns have been addressed, and the additional discussions with the reviewers have strengthened my positive opinion of the paper. For that, I will increase my score.

---

> > > ### Author Response · Authors · 2025-11-27
> > >
> > > We sincerely thank the reviewer for the positive feedback and for recognizing our additional analysis and clarifications. We appreciate the updated recommendation and your supportive assessment.

---

### Official Review · Reviewer_Yn6v · 2025-11-02

**Soundness:** 3
**Presentation:** 3
**Contribution:** 2
**Rating:** 4
**Confidence:** 4

**Summary:**

The paper introduces Sleep2Vec, a foundation model for nocturnal physiological recordings that learns a shared representation space across nine polysomnography related channels, including high rate neural and ocular signals and lower rate cardiorespiratory signals. The core idea is to treat concurrent nocturnal signals as complementary views of the same latent physiological state and to enforce alignment at the level of short epochs through a contrastive objective. The authors train on more than forty two thousand overnight recordings collected from several large public cohorts and harmonized through a common preprocessing pipeline. They propose a metadata aware contrastive loss named DASH InfoNCE that adjusts the weight of negatives by demographic attributes, recording site and subject night identity in order to reduce cohort shortcuts and to keep very similar samples from dominating the denominator. They report results on sleep staging for SHHS and WSC, on clinical outcome prediction for several conditions, and they include an ablation on leaving out individual modalities. They also attempt to characterize scaling behaviour along both the number of modalities and model capacity. The work aims to show that unified cross modal alignment for sleep becomes practical once enough heterogeneous data are brought together.

**Strengths:**

Strengths
The paper addresses an important practical problem in sleep medicine and mobile or low burden monitoring, which is the presence of many possible channel layouts and frequent missing sensors. Showing that one pre trained model can handle nine different signal types and remain robust when some are absent is a meaningful step toward realistic deployment across devices and centers. The methodological core is coherent. The use of two modality batches, masking, a single backbone and a shared projection head creates a simple recipe that other researchers can reproduce. The proposed contrastive objective is well argued. It uses age, gender and site to prioritize harder negatives, which is a clever way to turn epidemiological and acquisition metadata into useful training signals. The experimental section is quite extensive. The authors compare to specialized sleep staging networks on SHHS and show that the gap becomes small, sometimes negligible, despite the fact that their model was trained to be general and not tailored to sleep staging alone. They also evaluate on lower information channel groups such as IBI and respiratory signals, where the proposed model still outperforms prior foundation models and even reaches or exceeds specialized approaches in some metrics. The leave one out analysis is particularly useful since it quantifies the relative importance of each modality and confirms known clinical intuition that EEG and IBI carry more discriminative information for stage recognition than some of the other channels. The analysis of clinical prediction tasks with increasing numbers of modalities is also valuable as it suggests a predictable scaling trend.

**Weaknesses:**

Weaknesses
Although the paper claims better cross site generalization through metadata aware weighting, the current experiments do not fully isolate this effect. It would be more convincing to show a split in which one cohort is entirely held out during pre training and used only for evaluation, and to show that the gap between the standard InfoNCE and DASH InfoNCE enlarges in that setting. At present, the evidence comes from aggregate metrics and from the claim that site and demographic similarity are properly captured in the loss.
The paper positions itself as establishing scaling laws for nocturnal biosignals but the current analysis is still rather preliminary. The number of model sizes and modality subsets is limited and the curves are shown mostly for downstream performance without a deeper look at loss scaling during pre training, data efficiency or breakpoints where adding modalities stops helping. A more systematic study would make this part of the contribution stronger.
The reliance on a RoFormer backbone is sensible, yet the paper does not compare to other recent time series architectures that are specialized for long physiological sequences, for instance models with channel wise attention or structured state space models, which might be competitive or more efficient.
While the dataset section describes harmonization in detail, the paper does not quantify the amount of missing channels per cohort, nor does it show per cohort downstream results. Since the main motivation of the work is robustness to sensor dropout and to heterogeneous montages, a more explicit evaluation on realistic missing patterns would be helpful.
Finally, the work composes several ideas that have each appeared in some form in prior sleep foundation models, for example large scale contrastive pre training on ECG and respiratory signals and modality aware fusion. The novelty therefore resides mostly in the combination at scale and in the specific loss design rather than in a single transformative idea.

**Questions:**

1.	Can you report results in a leave one cohort out setting, for example training on HSP, SHHS, MrOS and MESA and evaluating only on WSC, with and without DASH InfoNCE, in order to show that the metadata aware weighting is particularly helpful when the target cohort is unseen.
2.	How sensitive is performance to the choice of the age kernel bandwidth and to the relative weights for same gender and same site samples. An ablation where these hyperparameters are perturbed would clarify to what extent the proposed loss needs tuning for new populations.
3.	In the two modality sampling scheme for pre training, have you tried curriculum strategies where the model first sees frequent and informative modality pairs such as EEG with respiratory effort and later progresses to rarer combinations. If so, did this accelerate convergence.
4.	For clinical outcome prediction, are the labels balanced across cohorts. If some conditions appear mostly in certain centers, the site aware part of the loss could risk encoding site related biases. Please clarify how this was mitigated.
5.	Could the authors provide a short comparison in compute cost between this approach and a variant that simply reconstructs masked modalities, since one of the selling points is that the alignment objective is more suitable for flexible inference.

---

> ### Author Response · Authors · 2025-11-20
> **Response  to Reviewer Yn6v  (Part 1)**
>
> We thank the reviewer for the detailed review and valuable comments, which have helped improve the clarity and quality of our work. Please find below the responses to each comment.
>
>
> # Weaknesses
> > "...It would be more convincing to show a split in which one cohort is entirely held out during pre training and used only for evaluation, and to show that the gap between the standard InfoNCE and DASH InfoNCE enlarges in that setting. At present, the evidence comes from aggregate metrics and from the claim that site and demographic similarity are properly captured in the loss..."
>
> We thank the reviewer for this suggestion.
> We have conducted evaluations on a new cohort APPLes using the models fine-tuned and evaluated on SHHS (Tab. 1 in the original manuscript) and WSC (Tab. 5 in the original manuscript), which is completely excluded from self-supervised pre-training and is only used for testing.
> We will make sure to incorporate this result in the revised manuscript.
>
> | Fine-tune dataset | Model                 | Overall Performance |       |       |       |       | Class-wise F1 |       |       |       |       |
> |-------------------|-----------------------|--------------------------|-------|-------|-------|-------|-------------------|-------|-------|-------|-------|
> |                   |                       | Acc.  | κ     | MF1  | Sens. | Spec. | wake  | n1    | n2    | n3    | rem   |
> | SHHS              | sleep2vec (InfoNCE)   | 0.768 | 0.667 | 0.635 | 0.733 | 0.934 | 0.905 | 0.240 | 0.795 | 0.355 | 0.881 |
> | SHHS              | sleep2vec             | 0.784 | 0.687 | 0.652 | 0.720 | 0.938 | 0.896 | 0.273 | 0.818 | 0.390 | 0.882 |
> | WSC               | sleep2vec (InfoNCE)   | 0.775 | 0.675 | 0.648 | 0.705 | 0.937 | 0.875 | 0.308 | 0.824 | 0.375 | 0.860 |
> | WSC               | sleep2vec             | 0.801 | 0.706 | 0.663 | 0.665 | 0.941 | 0.891 | 0.330 | 0.853 | 0.389 | 0.852 |
>
> > "...The number of model sizes and modality subsets is limited and the curves are shown mostly for downstream performance without a deeper look at loss scaling during pre training, data efficiency or breakpoints where adding modalities stops helping..."
>
>   We thank the reviewer for raising the point about scaling analysis. Our scaling claims are deliberately modest and are already supported along the model-size and modality axes in the original submission.
>   **From the perspective of model sizes,** Sec. 4.2.2 and Fig. 4 report results for three model sizes (63.5M, 133.7M, 238.2M parameters), spanning a 3.7× range and showing consistent, monotonic improvements with clear diminishing returns on both SHHS and WSC.
>   **On the modality side,** Sec. 4.2.3 and Fig. 5 further study modality diversity by varying the number of modalities from 1 to 9 and averaging over all combinations, again yielding smooth gains without signs of degradation as more modalities are added.
>   Together with the different channel subsets in Tab. 1 and 5 (EEG, IBI+Resp, EEG+EOG+EMG, full PSG), these results go beyond a single configuration and already cover multiple model capacities and a broad spectrum of modality subsets.
>   In the revised manuscript, we will further expand this analysis along the two axes the reviewer highlighted. First,  we will add these pre-training-level loss curves to the revised manuscript, **showing that the improvements observed on downstream tasks are mirrored at the pre-training level**. Second, we add new experiments that explicitly vary (i) the amount of pre-training data and (ii) the number of modalities used during pre-training, confirming that both downstream performance and pre-training loss improve smoothly as either data or modalities scale, with **no early "breakpoint" where adding modalities stops helping within the 9-channel PSG span**.
>
>   (i)-1 Performance on SHHS test set (Full PSG channels)
>   | Pre-training data |        Overall Performance on **SHHS**        |      |      |      |      |          Class-wise F1          |      |      |      |      |
> |-------------------|----------------------------------------|------|------|------|------|--------------------------------------|------|------|------|------|
> |                   | Acc. | κ    | MF1  | Sens. | Spec. | wake | n1   | n2   | n3   | rem  |
> | 25%               | 87.1 | 0.82 | 77.2 | 76.3 | 96.4 | 93.7 | 37.6 | 86.8 | 77.7 | 89.9 |
> | 50%               | 88.0 | 0.83 | 77.9 | 76.8 | 96.6 | 94.3 | 38.5 | 87.5 | 78.8 | 90.6 |
> | 75%               | 88.4 | 0.83 | 78.7 | 77.1 | 96.7 | 94.6 | 41.2 | 88.0 | 78.3 | 91.1 |
> | 100%              | 88.6 | 0.84 | 79.5 | 78.4 | 96.8 | 94.8 | 44.1 | 88.2 | 79.2 | 91.2 |

---

> ### Author Response · Authors · 2025-11-20
> **Response to Reviewer Yn6v (Part 2)**
>
> (i)-2 Performance on the APPLes cohort (Full PSG channels)
> | Pre-training data |        Overall Performance on **APPLes**        |      |      |      |      |  Class-wise f1          |      |      |      |      |
> |-------------------|----------------------------------------|------|------|------|------|--------------------------------------|------|------|------|------|
> |                   | Acc. | κ    | MF1  | Sens. | Spec. | wake | n1   | n2   | n3   | rem  |
> | 25%               | 76.1 | 0.65 | 62.7 | 70.9 | 93.1 | 88.5 | 23.1 | 79.5 | 36.2 | 86.4 |
> | 50%               | 76.8 | 0.66 | 63.0 | 71.7 | 93.3 | 89.5 | 21.6 | 80.0 | 36.2 | 87.7 |
> | 75%               | 78.0 | 0.68 | 64.6 | 71.7 | 93.6 | 89.1 | 25.4 | 81.5 | 38.8 | 88.0 |
> | 100%              | 78.4 | 0.69 | 65.2 | 72.0 | 93.8 | 89.6 | 27.3 | 81.8 | 39.0 | 88.2 |
>
> (ii)-1 Performance on the SHHS test set
> | channels      | Curriculum   |        Overall Performance        |      |      |      |      |          Class-wise F1          |      |      |      |      |
> |---------------|--------------|----------------------------------------|------|------|------|------|--------------------------------------|------|------|------|------|
> |               |              | Acc. | κ    | MF1  | Sens. | Spec. | wake | n1   | n2   | n3   | rem  |
> | EEG       | Stage 1      | 86.9 | 0.81 | 77.4 | 76.3 | 96.3 | 93.5 | 41.0 | 86.7 | 77.2 | 88.6 |
> |               | Stage 2      | 87.0 | 0.82 | 77.7 | 76.5 | 96.3 | 93.6 | 42.5 | 86.7 | 76.8 | 88.8 |
> |               | Stage 3      | 87.3 | 0.82 | 77.2 | 75.7 | 96.4 | 93.7 | 38.9 | 87.0 | 77.6 | 88.9 |
> |               | Single-stage | 87.4 | 0.82 | 77.3 | 76.6 | 96.5 | 94.2 | 40.1 | 86.5 | 77.7 | 88.3 |
> | RESP+IBI | Stage 1      | 82.1 | 0.74 | 67.5 | 66.6 | 94.8 | 91.9 | 15.2 | 80.5 | 63.7 | 86.3 |
> |               | Stage 2      | 82.2 | 0.75 | 66.3 | 66.9 | 94.9 | 91.8 | 6.6  | 80.2 | 66.2 | 86.6 |
> |               | Stage 3      | 82.5 | 0.75 | 68.2 | 67.3 | 95.0 | 92.3 | 17.1 | 80.9 | 63.7 | 86.8 |
> |               | Single-stage | 83.0 | 0.75 | 65.9 | 65.8 | 95.1 | 86.6 | 5.3  | 80.9 | 64.3 | 86.6 |
>
> (ii)-2 Performance on the APPLes cohort
> | channels      | Curriculum   |        Overall Performance        |      |      |      |      |          Class-wise f1          |      |      |      |      |
> |---------------|--------------|----------------------------------------|------|------|------|------|--------------------------------------|------|------|------|------|
> |               |              | Acc. | κ    | MF1  | Sens. | Spec. | wake | n1   | n2   | n3   | rem  |
> | EEG       | Stage 1      | 76.6 | 0.66 | 63.6 | 72.8 | 93.3 | 89.4 | 25.4 | 79.8 | 36.7 | 86.5 |
> |               | Stage 2      | 77.4 | 0.67 | 64.3 | 72.5 | 93.5 | 89.6 | 26.8 | 80.8 | 37.6 | 86.9 |
> |               | Stage 3      | 77.2 | 0.67 | 63.2 | 72.1 | 93.4 | 89.1 | 20.8 | 80.8 | 37.8 | 87.3 |
> |               | Single-stage | 76.7 | 0.66 | 62.5 | 71.9 | 93.3 | 89.6 | 19.1 | 80.0 | 37.3 | 86.7 |
> | RESP+IBI | Stage 1      | 72.3 | 0.60 | 56.8 | 66.4 | 91.9 | 86.7 | 7.1  | 75.3 | 31.0 | 83.8 |
> |               | Stage 2      | 71.2 | 0.59 | 55.4 | 66.7 | 91.7 | 86.8 | 3.1  | 73.9 | 29.0 | 83.9 |
> |               | Stage 3      | 72.3 | 0.60 | 57.1 | 67.0 | 91.9 | 87.6 | 7.8  | 74.8 | 30.1 | 85.0 |
> |               | Single-stage | 73.2 | 0.61 | 57.8 | 66.2 | 92.1 | 86.5 | 10.2 | 76.5 | 31.5 | 84.2 |
>
>   Note that Single-stage model denotes the proposed DASH-InfoNCE sleep2vec model. Stage 1 trains the model only on the most frequent and informative pairs (EEG + IBI + RESP) until convergence. Stage 2 introduces the second group of modalities (EOG, ECG and Nasal airflow) and continued the training from the Stage 1 checkpoint. Stage 3 adds the remaining, less common modalities (EMG, ABD/Thor and SpO₂), again continuing from the Stage 2 checkpoint.
>
>   While a fully exhaustive scaling study over many more model sizes and amount of data would be computationally prohibitive given that each run uses more than 42,000 nights and 9 modalities, the existing and newly added results consistently support our central claim: within the practically relevant range we study, sleep2vec exhibits predictable, well-behaved scaling with respect to model size, data volume, and modality diversity.

---

> ### Author Response · Authors · 2025-11-20
> **Response to Reviewer Yn6v (Part 3)**
>
> > "...The reliance on a RoFormer backbone is sensible, yet the paper does not compare to other recent time series architectures that are specialized for long physiological sequences, for instance models with channel wise attention or structured state space models, which might be competitive or more efficient..."
>
> We appreciate the reviewer's suggestion to compare against more recent time series architectures such as channel-wise attention models and structured state space models.
>
> We would like to emphasize that the goal in this work is not to argue that RoFormer is the uniquely optimal backbone for PSG, but to show that large-scale multimodal pre-training with DASH-InfoNCE yields representations that are competitive with strong, task-specific sequence models. To this end, our current experiments already compare against several long-sequence PSG architectures with sophisticated temporal modeling (e.g., L-SeqSleepNet, SleepTransformer, PFTSleep) on both SHHS and WSC, and sleep2vec consistently matches or outperforms them across multiple channel configurations. For an apples-to-apples comparison, we focused on baselines that report results on SHHS and WSC (or release code that can be reliably adapted to these settings). Many recent channel-wise attention and SSM-based models either target different domains or datasets, or lack publicly available implementations, which makes a fair quantitative comparison challenging within the rebuttal timeline.
>
> In the revision, we will (i) expand the related-work section to explicitly discuss these model families and their relation to our framework, and (ii) where feasible, add results for a representative SSM / channel-wise attention baseline re-implemented on SHHS and WSC.   We would also be happy to incorporate any specific baselines the reviewer has in mind, provided code and data are available for a fair evaluation.
>
> > "...The paper does not quantify the amount of missing channels per cohort, nor does it show per cohort downstream results. A more explicit evaluation on realistic missing patterns would be helpful..."
>
> We thank the reviewer for the insightful comments regarding missing channels and realistic missing patterns.
> **Our primary motivation in this work is not robustness to arbitrary, per-night sensor dropouts, but to learn a unified physiological representation on full PSG that can be reliably adapted to different fixed sensor configurations and devices (e.g., full PSG, EEG-only, IBI+Resp, EEG+EOG+EMG).** The clinical PSG cohorts we use are collected under standardized sleep-lab protocols with high data quality and stringent preprocessing, so severe night-level **sensor dropout is rare and such records are typically excluded**.
> In this setting, the realistic "missing patterns" are therefore driven mainly by protocol-level differences in which channels are available across cohorts, rather than random loss of sensors during a given night. Our current experiments already target this scenario: we pre-train once on full PSG and then evaluate downstream performance across multiple datasets and channel subsets, showing that the learned representation transfers robustly to diverse sensor combinations. In the revised manuscript, we will clarify this setting more explicitly and provide a more detailed description of the channel configurations of each dataset and the type of channel-configuration robustness that our evaluation focuses on.

---

> ### Author Response · Authors · 2025-11-20
> **Response to Reviewer Yn6v (Part 4)**
>
> > "...the work composes several ideas that have each appeared in some form in prior sleep foundation models, for example large scale contrastive pre training on ECG and respiratory signals and modality aware fusion. The novelty therefore resides mostly in the combination at scale and in the specific loss design rather than in a single transformative idea..."
>
> We appreciate the reviewer's careful assessment of novelty and the pointer to prior sleep foundation models. We agree that our work builds on existing ingredients such as contrastive pre-training and multimodal fusion, but we believe the contribution goes beyond a mere combination of known ideas.
>
> First, prior sleep FMs typically target either (i) a single or very small set of modalities (e.g., ECG and Respiration) or (ii) a fixed PSG montage for a specific task. **In contrast, sleep2vec is, to our knowledge, the first framework that performs time-step cross-modal alignment across nine PSG modalities (EEG, EOG, EMG, ECG, airflow, belts, SpO₂, IBI, respiratory effect) jointly collected from five cohorts and 42,249 full-night studies**, and is explicitly designed so that the same pre-trained encoder can be fine-tuned on arbitrary modality subsets (EEG, IBI+Resp, EEG+EOG+EMG, full PSG) without re-training separate FMs for each configuration. This capability is central to our goal of "arbitrary modality inference" and is not supported by existing ECG/respiratory-only or single-montage sleep FMs.
>
> Second, while contrastive learning has indeed been used for physiological signals, **our Demography, Age, Site & History-aware InfoNCE (DASH-InfoNCE) is not a minor variant**: it restructures the negative set using demographic and acquisition metadata and modulates same-night pseudo-negatives, specifically to mitigate cohort-specific shortcuts and improve cross-center generalization. Ablations in Tab. 1 and 5 and Fig. 5 show that DASH-InfoNCE consistently and sometimes substantially outperforms standard InfoNCE across channel subsets and clinical prediction tasks, indicating that the loss design contributes genuine modelling gains rather than cosmetic differences.
>
> Third, we provide, to our knowledge, the first systematic scaling-law study for PSG foundation models along both the parameter and modality-diversity axes (Fig. 4 and Fig. 5), demonstrating predictable improvements with larger models and more modalities on SHHS and WSC.
>
> Finally, sleep2vec is evaluated as a single pre-trained model across two datasets, multiple channel configurations, and both staging and disease tasks, often matching or surpassing specialized long-sequence PSG architectures and prior FMs, while they are typically pre-trained per-configuration.
>
> Taken together, we view the core novelty of our submission as (i) a unified cross-modal PSG foundation model capable of robust inference from arbitrary modality subsets across cohorts, and (ii) a metadata-aware contrastive objective with accompanying scaling analysis, rather than a loose aggregation of previously known techniques.
>
> # Questions
>
> > 1. Can you report results in a leave one cohort out setting, for example training on HSP, SHHS, MrOS and MESA and evaluating only on WSC, with and without DASH InfoNCE, in order to show that the metadata aware weighting is particularly helpful when the target cohort is unseen.
>
> We have conducted evaluations on a new cohort APPLes using the models fine-tuned and evaluated on SHHS (Tab. 1 in the original manuscript) and WSC (Tab. 5 in the original manuscript), which is completely excluded from self-supervised pre-training and is only used for testing.
> We will make sure to incorporate this result in the revised manuscript.
>
> | Fine-tune dataset | Model                 | Overall Performance |       |       |       |       | Class-wise F1 |       |       |       |       |
> |-------------------|-----------------------|--------------------------|-------|-------|-------|-------|-------------------|-------|-------|-------|-------|
> |                   |                       | Acc.  | κ     | MF1  | Sens. | Spec. | wake  | n1    | n2    | n3    | rem   |
> | SHHS              | sleep2vec (InfoNCE)   | 0.768 | 0.667 | 0.635 | 0.733 | 0.934 | 0.905 | 0.240 | 0.795 | 0.355 | 0.881 |
> | SHHS              | sleep2vec             | 0.784 | 0.687 | 0.652 | 0.720 | 0.938 | 0.896 | 0.273 | 0.818 | 0.390 | 0.882 |
> | WSC               | sleep2vec (InfoNCE)   | 0.775 | 0.675 | 0.648 | 0.705 | 0.937 | 0.875 | 0.308 | 0.824 | 0.375 | 0.860 |
> | WSC               | sleep2vec             | 0.801 | 0.706 | 0.663 | 0.665 | 0.941 | 0.891 | 0.330 | 0.853 | 0.389 | 0.852 |

---

> ### Author Response · Authors · 2025-11-20
> **Response to Reviewer Yn6v (Part 5)**
>
> > 2. How sensitive is performance to the choice of the age kernel bandwidth and to the relative weights for same gender and same site samples. An ablation where these hyperparameters are perturbed would clarify to what extent the proposed loss needs tuning for new populations.
>
> We appreciate the reviewer's question regarding the sensitivity of our method to the metadata-weighting hyperparameters. To assess this, we performed a study in which we perturbed the three key components of the weighting scheme: the cross-center penalty, the gender penalty factor, and the age-kernel bandwidth. For each parameter, we varied its value over a broad range around the default (0.8 for both penalties and 20 for the age kernel), namely 0.6, 0.7, 0.8, 0.9 for the gender and cross-center penalties and 5, 10, 20, 30 for the age-kernel width.
>
> **The resulting pre-training loss landscape will be added to the revised manuscript.** All metadata-aware surfaces are nearly flat and tightly clustered, indicating that the loss varies only marginally when these hyperparameters are perturbed. The variation across configurations is small compared to the gap between vanilla InfoNCE and our metadata-aware variants, suggesting that the proposed loss is not overly sensitive to moderate changes in these hyperparameters and should not require careful retuning when applied to new cohorts.
>
> > 3. In the two modality sampling scheme for pre training, have you tried curriculum strategies where the model first sees frequent and informative modality pairs such as EEG with respiratory effort and later progresses to rarer combinations. If so, did this accelerate convergence.
>
>   We thank the reviewer for the thoughtful suggestion. To investigate whether a curriculum over modality pairs can benefit pre-training, a three-stage experiment in which the model progressively encounters increasingly rare modality combinations was conducted.
>   In Stage 1, we trained the model only on the most frequent and informative pairs (EEG + IBI + respiratory effort) until convergence. In Stage 2, we introduced the second group of modalities (EOG, ECG and Nasal airflow) and continued training from the Stage 1 checkpoint. Stage 3 then added the remaining, less common modalities (EMG, ABD/Thor, SpO₂), again continuing from the Stage 2 checkpoint.
>   We then evaluated the resulting models on downstream sleep staging. While the curriculum schedule produced models with stable within-cohort performance, it did not yield clear improvements on external cohorts, and we observed slight fluctuations compared to the baseline.
>
> Performance on the SHHS test set
> | channels      | Curriculum   |        Overall Performance        |      |      |      |      |          Class-wise F1          |      |      |      |      |
> |---------------|--------------|----------------------------------------|------|------|------|------|--------------------------------------|------|------|------|------|
> |               |              | Acc. | κ    | MF1  | Sens. | Spec. | wake | n1   | n2   | n3   | rem  |
> | EEG       | Stage 1      | 86.9 | 0.81 | 77.4 | 76.3 | 96.3 | 93.5 | 41.0 | 86.7 | 77.2 | 88.6 |
> |               | Stage 2      | 87.0 | 0.82 | 77.7 | 76.5 | 96.3 | 93.6 | 42.5 | 86.7 | 76.8 | 88.8 |
> |               | Stage 3      | 87.3 | 0.82 | 77.2 | 75.7 | 96.4 | 93.7 | 38.9 | 87.0 | 77.6 | 88.9 |
> |               | Single-stage | 87.4 | 0.82 | 77.3 | 76.6 | 96.5 | 94.2 | 40.1 | 86.5 | 77.7 | 88.3 |
> | RESP+IBI | Stage 1      | 82.1 | 0.74 | 67.5 | 66.6 | 94.8 | 91.9 | 15.2 | 80.5 | 63.7 | 86.3 |
> |               | Stage 2      | 82.2 | 0.75 | 66.3 | 66.9 | 94.9 | 91.8 | 6.6  | 80.2 | 66.2 | 86.6 |
> |               | Stage 3      | 82.5 | 0.75 | 68.2 | 67.3 | 95.0 | 92.3 | 17.1 | 80.9 | 63.7 | 86.8 |
> |               | Single-stage | 83.0 | 0.75 | 65.9 | 65.8 | 95.1 | 86.6 | 5.3  | 80.9 | 64.3 | 86.6 |

---

> ### Author Response · Authors · 2025-11-20
> **Response to Reviewer Yn6v (Part 6)**
>
> Performance on the APPLes cohort
> | channels      | Curriculum   |        Overall Performance        |      |      |      |      |          Class-wise f1          |      |      |      |      |
> |---------------|--------------|----------------------------------------|------|------|------|------|--------------------------------------|------|------|------|------|
> |               |              | Acc. | κ    | MF1  | Sens. | Spec. | wake | n1   | n2   | n3   | rem  |
> | EEG       | Stage 1      | 76.6 | 0.66 | 63.6 | 72.8 | 93.3 | 89.4 | 25.4 | 79.8 | 36.7 | 86.5 |
> |               | Stage 2      | 77.4 | 0.67 | 64.3 | 72.5 | 93.5 | 89.6 | 26.8 | 80.8 | 37.6 | 86.9 |
> |               | Stage 3      | 77.2 | 0.67 | 63.2 | 72.1 | 93.4 | 89.1 | 20.8 | 80.8 | 37.8 | 87.3 |
> |               | Single-stage | 76.7 | 0.66 | 62.5 | 71.9 | 93.3 | 89.6 | 19.1 | 80.0 | 37.3 | 86.7 |
> | RESP+IBI | Stage 1      | 72.3 | 0.60 | 56.8 | 66.4 | 91.9 | 86.7 | 7.1  | 75.3 | 31.0 | 83.8 |
> |               | Stage 2      | 71.2 | 0.59 | 55.4 | 66.7 | 91.7 | 86.8 | 3.1  | 73.9 | 29.0 | 83.9 |
> |               | Stage 3      | 72.3 | 0.60 | 57.1 | 67.0 | 91.9 | 87.6 | 7.8  | 74.8 | 30.1 | 85.0 |
> |               | Single-stage | 73.2 | 0.61 | 57.8 | 66.2 | 92.1 | 86.5 | 10.2 | 76.5 | 31.5 | 84.2 |
>
> Note that Single-stage model denotes the proposed DASH-InfoNCE sleep2vec model.
>
> > 4. For clinical outcome prediction, are the labels balanced across cohorts. If some conditions appear mostly in certain centers, the site aware part of the loss could risk encoding site related biases. Please clarify how this was mitigated.
>
> We thank the reviewer for raising this important point. We would like to clarify that all clinical outcome prediction experiments in the current paper are performed within a single cohort (SHHS) rather than by pooling labels across multiple cohorts. As described in Sec. 3.1, clinical diagnosis labels are only available for SHHS, and downstream splits for these tasks are created at the participant level within SHHS using the same train/validation/test partition as for pre-training. Thus, there is no cross-cohort label imbalance for the clinical prediction results reported in Fig. 5; prevalence differences between HSP, MrOS, MESA, and WSC do not directly enter the supervised clinical experiments.
>
> Regarding the site-aware component of DASH-InfoNCE, we emphasize that site metadata is used only during pre-training to weight negative samples and is never provided as an input feature or label in any downstream classifier. In particular, negatives from the same site receive higher weights in the denominator of the contrastive loss. This design intentionally makes any representation that strongly encodes site-specific cues harder to optimize: if two segments are similar mainly because they come from the same center (e.g., due to device or scoring idiosyncrasies), they will appear as high-weight hard negatives and the loss will penalize such shortcuts. **Coupled with the standardized preprocessing and harmonization pipeline we apply across all cohorts to minimize center-specific artifacts, the site-aware weighting is therefore aimed at suppressing, rather than amplifying, site-related biases.**
>
> We will clarify in the revised manuscript that (i) clinical outcome prediction is conducted within SHHS only, (ii) site information is never used as a downstream covariate, and (iii) the role of the site-aware weighting in DASH-InfoNCE is to discourage the model from relying on site-specific shortcuts, thereby improving cross-center robustness rather than encoding spurious site-label associations.

---

> ### Author Response · Authors · 2025-11-20
> **Response to Reviewer Yn6v (Part 7)**
>
> > 5. Could the authors provide a short comparison in compute cost between this approach and a variant that simply reconstructs masked modalities, since one of the selling points is that the alignment objective is more suitable for flexible inference.
>
> We thank the reviewer for this suggestion. To clarify, **our notion of "flexible inference" refers to the ability to operate on arbitrary subsets of available sensors without retraining or explicit imputation, rather than to a reduction in FLOPs compared with reconstruction-based objectives**. This is the sense in which we emphasize robustness to arbitrary modality subsets and sensor dropout. Our claim is that an alignment objective is architecturally aligned with this goal, because it directly maps whatever sensors are present into a shared representation space.
>
> From a compute perspective, both approaches share the same backbone encoders. Our alignment objective adds only lightweight linear projection heads and an in-batch similarity matrix of size $B \times B$, so the encoder forward/backward pass remains the dominant cost. In contrast, a masked-modality reconstruction variant requires per-modality decoders that regress dense waveforms for masked channels. **The cost of these decoders scales with the output sequence length and the number of modalities, and for PSG-length windows this typically dominates both FLOPs and activation memory**. At inference time, alignment performs no decoding: representations are formed directly from whatever subset of sensors is present, so latency and memory scale roughly linearly with the number of observed channels. A reconstruction-based system either (i) runs decoders to impute missing channels and then re-encodes them, adding extra compute and memory, or (ii) accepts a mismatch between the training objective (reconstruction) and its test-time usage under sensor dropout. This is why we argue that alignment is better suited to flexible inference in our setting. It avoids per-modality decoders, keeps inference simple, and scales cleanly with variable sensor availability.
>
> In addition, we implemented a cross-modal reconstruction baseline (shared encoder plus per-modality decoders) under the same backbone and data pipeline. Besides being more expensive to train, it did not converge reliably, exhibiting unstable training despite extensive tuning of the optimizer, learning rate, and masking ratio. We attribute this to the difficulty of regressing long, heterogeneous physiological waveforms at high temporal resolution, which further underscores the practicality of an alignment objective for PSG-scale pre-training.

---

> > ### Comment · Reviewer_Yn6v · 2025-11-28
> > **Thanks for your reply.**
> >
> > Thank you for your reply. The authors have already solved all of my problems. The score should be raised to 6.

---

### Author Response · Authors · 2025-11-30
**Clarification of Addressed Issues and Final Reviewer Assessments Pre-Rollback (Part 1)**

**Author summary after score rollback**

We understand that, due to the recent incident, reviewer scores have been reverted. However, after our rebuttal and additional experiments, **three of the four reviewers explicitly stated that their concerns had been resolved**, and **two updated their scores to 8 and recommended acceptance**, while a **third asked to raise their score to 6**. These statements remain visible in the discussion thread but are not reflected in the rolled‑back scores.

Below we (i) summarise the main changes made to the paper in response to the reviews, and (ii) state the status of each review at the time the system stopped allowing new posts.

---

## Main changes to the paper

Across all reviewers we made the following substantive additions and clarifications (now reflected in the latest revision and in our author responses).

* **New cross‑cohort evaluation (APPLES):**
  We added experiments on the APPLES cohort, which is completely excluded from self‑supervised pre‑training and only used for testing. Using models fine‑tuned on SHHS and WSC, we show that the proposed DASH‑InfoNCE consistently improves both in‑cohort and APPLes performance over vanilla InfoNCE, directly addressing concerns about cross‑site generalisation (R_Yn6v).

* **Scaling with data and modalities:**
  Beyond model‑size scaling, we now systematically vary the *amount of pre‑training data* (25/50/75/100%) and report monotonic improvements on both SHHS and APPLES, together with modality‑diversity scaling (1–9 modalities). This strengthens the scaling‑law claims (R_Yn6v, R_7ymV).

* **Metadata ablations and sensitivity:**
  We added ablations activating each metadata branch (age, gender, site) separately and report their individual contributions to retrieval and downstream performance, as well as sensitivity analyses over the age‑kernel bandwidth and gender/site weights. These show that each branch helps and that DASH‑InfoNCE is not heavily hyperparameter‑sensitive (R_Yn6v, R_BLg5).

* **Curriculum vs single‑stage pre‑training:**
  We implemented the proposed curriculum over modality pairs (EEG+IBI+Resp → +EOG/ECG/airflow → +EMG/ABD/Thor/SpO₂). Results show no clear advantage over our simpler single‑stage training, justifying the simpler setup (R_Yn6v).

* **Fusion strategy ablation (Concat / Mean / Gating):**
  We compared the three fusion schemes and found that Gating yields the best performance with modest gains over Mean, while Concat is more expensive without clear benefit. This justifies the chosen fusion module and shows its interpretability comes at negligible cost (R_7LYS).

* **Masking‑rate ablation:**
  We added experiments with 0%, 15%, and 30% masking. A moderate 15% masking rate improves both within‑cohort and cross‑cohort performance compared to no masking or 30%, supporting our default choice (R_7ymV).

* **Clarified data statistics and splits:**
  We now provide detailed statistics for all cohorts (age ranges, mean±SD duration, total hours, number of subjects and recordings) and clearly state that *all* downstream results use subject‑level 8:1:1 splits with no participant overlap between pre‑training, fine‑tuning, and test. This clarifies subject‑independent evaluation and dataset scale (R_7ymV, R_BLg5).

* **Clarified novelty, baselines, and training setup:**
  We expanded the related‑work discussion to situate sleep2vec relative to CIMSleepNet, MultiConsSleepNet, SleepSMC, SleepFM, SleepFounder, and PFTSleep, and added an appendix describing how we instantiate and pre‑train each FM baseline on our splits. We also clarified that sleep2vec uses *only* contrastive alignment (no reconstruction objective) and that baselines are trained using their original protocols on the same subject splits and channel subsets (R_7LYS, R_BLg5).

---

> ### Author Response · Authors · 2025-11-30
> **Clarification of Addressed Issues and Final Reviewer Assessments Pre-Rollback (Part 2)**
>
> ## Reviewer‑by‑reviewer status at discussion freeze
>
> ### Reviewer 7ymV  *(original score: 4)*
>
> **Main concerns (now addressed)**
>
> * Lack of ablations on pre‑training variants (e.g. masking rate, alternative objectives).
> * Incomplete scaling‑law analysis (only model size, no data‑size scaling).
> * Incomplete data statistics; unclear whether heterogeneous‑input capability comes from backbone vs. loss.
> * Clarifications on which cohorts are used for which tasks and details of fine‑tuning.
>
> **Changes made**
>
> * Added masking‑rate ablations (0/15/30%) showing 15% masking gives the best in‑ and out‑of‑cohort performance.
> * Added data‑scaling experiments (25/50/75/100% pre‑training data).
> * Added a detailed data‑statistics table and clarified that all downstream evaluation is subject‑independent.
> * Clarified that heterogeneous‑input handling comes from modality‑specific tokenizers plus a modality‑agnostic backbone, while DASH‑InfoNCE improves cross‑cohort robustness on top of that.
> * Summarised and standardised the fine‑tuning protocols across all baselines.
>
> **Reviewer’s final comment before freeze**
> In a comment dated **25 Nov**, R_7ymV states that all concerns have been *fully addressed*, praises the new ablations and data statistics, and writes that they “recommend its acceptance, as reflected in my updated rating.”
>
> We then acknowledged that they had **raised the score to 8** in our author reply on 26 Nov, which is also visible on the forum.
>
> > **Our belief:** R_7ymV’s final stance is clearly **accept with score 8**, despite the displayed rating having reverted to 4.
>
> ---
>
> ### Reviewer 7LYS  *(original score: 6)*
>
> **Main concerns (now addressed)**
>
> * Potential confusion around “intra‑subject / inter‑subject” segments in Fig. 2.
> * Lack of clear positioning of SleepFM/PFTSleep and other FM baselines.
> * Questions about the compute trade‑off of full channels vs EEG‑only and of FM vs specialised models.
> * Lack of explicit comparison of fusion strategies and discussion of where scaling plateaus.
>
> **Changes made**
>
> * Clarified that “intra/inter‑subject segments” in Fig. 2 are descriptive labels tied directly to the DASH‑InfoNCE formulation, not a separate mechanism.
> * Expanded related work and added an appendix subsection describing baseline FMs, their objectives, and how we re‑train them on our splits.
> * Explained compute scaling (roughly linear in #modalities) and that pre‑training cost is amortised over many tasks and montages; clarified intent of full‑channel results.
> * Added the fusion ablation (Concat vs Mean vs Gating) showing Gating slightly outperforms Mean and is more efficient than Concat.
> * Discussed that within our 63M–238M parameter range we see monotonic but diminishing returns, with no clear saturation.
>
> **Reviewer’s final comment before freeze**
> In a comment dated **27 Nov**, R_7LYS writes that “all my concerns have been addressed,” that the additional experiments “have strengthened my positive opinion of the paper,” and that they “will increase my score.”
>
> Our subsequent acknowledgement explicitly thanks them **“for raising the score to 8”**, which is recorded in our 28 Nov author comment.
>
> > **Our belief:** R_7LYS’s final stance is **positive, with score 8**, although the displayed rating has reverted to 6.
>
> ---
>
> ### Reviewer Yn6v  *(original score: 4)*
>
> **Main concerns (now addressed)**
>
> * Need for a cohort‑held‑out evaluation to demonstrate cross‑site generalisation and the benefit of DASH‑InfoNCE vs vanilla InfoNCE.
> * More systematic scaling analysis (data efficiency, loss scaling).
> * Comparison to other long‑sequence architectures; better characterisation of missing‑channel patterns.
> * Clarification of how site metadata affects potential biases; compute vs reconstruction‑based variants.
>
> **Changes made**
>
> * Added APPLes cohort experiments (unseen in pre‑training) using models fine‑tuned on SHHS/WSC, showing consistent gains of DASH‑InfoNCE over InfoNCE on both in‑cohort and APPLes performance.
> * Added data‑scaling experiments and pre‑training‑loss curves; added curriculum vs single‑stage experiments.
> * Clarified that site metadata is used only in the contrastive loss (never as downstream input) and is designed to *discourage* site shortcuts.
> * Explained why a reconstruction objective is less suitable in our setting and summarised unsuccessful attempts at cross‑modal reconstruction under the same compute budget.
>
> **Reviewer’s final comment before freeze**
> In a **28 Nov** comment replying to our Part‑7 response, R_Yn6v writes:
>
> > “The authors have already solved all of my problems. **The score should be raised to 6.**”
>
> > **Our belief:** R_Yn6v’s final stance is **marginally positive (score 6)**, although the rating field has reverted to 4.

---

> > ### Author Response · Authors · 2025-11-30
> > **Clarification of Addressed Issues and Final Reviewer Assessments Pre-Rollback (Part 3)**
> >
> > ### Reviewer BLg5  *(original score: 2)*
> >
> > **Main concerns**
> >
> > * Perceived lack of originality relative to CIMSleepNet, MultiConsSleepNet, SleepSMC.
> > * “Outdated” baselines and absence of certain recent models.
> > * Lack of explicit ablations for each module in the loss; unclear robustness under missing modalities.
> > * Claimed absence of cross‑subject evaluation and statistical‑significance analysis; ambiguity in Table‑1 settings.
> >
> > **Changes made**
> >
> > * Clarified conceptual and experimental differences from CIMSleepNet / MultiConsSleepNet / SleepSMC, particularly in scope (nine modalities across five large cohorts, metadata‑aware loss, clinical prediction and scaling laws).
> > * Emphasised that we already compare with recent PSG FMs SleepFM, SleepFounder, and PFTSleep retrained on our splits, in addition to canonical staging baselines.
> > * Added detailed ablations of age/gender/site branches and a cross‑modal retrieval metric to quantify alignment; clarified that sleep2vec uses contrastive alignment only (no reconstruction), with reconstruction baselines found unstable under comparable budgets.
> > * Explicitly documented subject‑independent splits and clarified that the first column in Tables 1 and 5 denotes *inference‑time* channel subsets.
> > * Committed to reporting mean±std over multiple fine‑tuning runs for key configurations in the camera‑ready.
> >
> > **Status**
> > R_BLg5 has not posted any follow‑up comments after our responses, and their displayed rating remains 2. We therefore cannot know whether their opinion changed, though many of the issues they raised (cross‑subject evaluation, missing ablations, clarity of tables) are now directly addressed in the revised manuscript and rebuttal.
> >
> > > **Our belief:** R_BLg5 is the only reviewer for whom we do not have an explicit post‑discussion update; their score likely remains 2.
> >
> > ## Overall picture at time of freeze
> >
> > Putting this together:
> >
> > * **Pre‑discussion scores (now shown after rollback):** 2 (BLg5), 4 (Yn6v), 4 (7ymV), 6 (7LYS).
> > * **Stated post‑discussion positions before the freeze (based on visible comments):**
> >
> >   * R_7ymV – “recommend its acceptance… updated rating” (score raised to 8).
> >   * R_7LYS – “all my concerns have been addressed… I will increase my score” (raised to 8).
> >   * R_Yn6v – “The authors have already solved all of my problems. The score should be raised to 6.”
> >   * R_BLg5 – no follow‑up comment; rating appears unchanged at 2.
> >
> > Thus, **three of four reviewers indicated improved assessments, with two strong accepts (8, 8) and one positive borderline (6)**, based on the revised paper and additional experiments described above. Only one reviewer (BLg5) has not updated their opinion publicly.
> >
> > We fully understand the need to revert scores for fairness across submissions. Our goal with this summary is simply to ensure that the AC can efficiently see the state of the discussion *at the time it was frozen* and the substantive changes made to address reviewers’ concerns, rather than relying solely on the pre‑discussion scores.

---

### Meta-Review · Area_Chair_1kqT · 2026-01-04

**Summary:**

The paper introduces Sleep2Vec, a foundation model for nocturnal physiological signals. It learns a shared representation across nine PSG-related channels, covering both high-rate signals like EEG and EOG, and lower-rate cardiorespiratory signals. The idea and formulation are clear and interesting, and all reviewers acknowledge this as a strong point.

The experimental section is thorough. The authors evaluate on several well-known datasets, which are properly cited, and include useful analysis when increasing the number of modalities. The model also shows robust behavior when some signals are missing, which is an important practical advantage.

During the rebuttal, the authors made a very strong effort to address reviewer concerns. They added leave-one-cohort-out evaluations, sensitivity analysis for kernel bandwidth, clear ablations on feature fusion methods (concatenation, averaging, gating), and analysis of performance versus model size. They also added ablations on masking rate, showing stable performance across settings, and scaling experiments with respect to data size, which better support the scaling claims. In addition, they included detailed dataset statistics, improving clarity and transparency.

Three out of four reviewers indicated acceptance after seeing these new results. Reviewer BLg5 remained critical, but the authors’ responses and new ablation studies clearly move the paper in the direction suggested by this reviewer. Many of the added clarifications also address concerns raised by BLg5 indirectly.

One important point is that all new experiments and clarifications from the rebuttal must be fully integrated into the final version of the paper. In particular, the relation to the three references mentioned by BLg5 should be clearly discussed, and a comparison with MultiConsSleepNet and SleepSMC would further strengthen the work.

Overall, given two strong accept scores, one weak accept, and the solid rebuttal addressing most concerns, I recommend acceptance.

**Reviewer Concerns:**

**Addressed by the rebuttal**
- Added leave-one-cohort-out analysis to test generalization across cohorts.
- Studied sensitivity to kernel bandwidth in the DASH-InfoNCE objective.
- Clarified feature fusion strategies (concatenation, averaging, gating).
- Added analysis of performance versus model size and number of parameters.
- Added ablations on masking rate in pretraining.
- Added scaling experiments with respect to data size, with detailed results.
- Included detailed dataset statistics for better clarity and transparency.
- Clarified robustness when some modalities are missing.

**Still outstanding**
- New experiments and analyses from the rebuttal are not yet integrated into the main paper.
- The relation to prior work highlighted by reviewer BLg5 (three specific references) needs clearer discussion.
- No direct comparison with MultiConsSleepNet and SleepSMC models.
- Some concerns from reviewer BLg5 about positioning with respect to existing sleep foundation models remain partially open.

**Reviewer Scores:**

- **Reviewer 7LYS**
  Acknowledged an increase from 6 to 8.

- **Reviewer Yn6v**
  Acknowledged an increase from 4 to 6.

- **Reviewer BLg5**
  Initial score was 2. Likely to raise to at least 4 after the rebuttal.

- **Reviewer 7ymV**
  Acknowledged an increase from 4 to 8.

Average rebuttal score is 6-6.5

---

### Decision · Program_Chairs · 2026-01-26

Accept (Poster)